# Heuristic algorithms in evolutionary computation and modular organization of biological macromolecules: Applications to *in vitro* evolution

**Alexander V. Spirov** [1,2]*, **Ekaterina M. Myasnikova** [3]

1 I. M. Sechenov Institute of Evolutionary Physiology and Biochemistry Russian Academy of Sciences, St. Petersburg, Russia, 2 The Institute of Scientific Information for Social Sciences RAS, Moscow, Russia, 3 Peter the Great St. Petersburg Polytechnic University, St. Petersburg, Russia

* alexander.spirov@gmail.com

**Data Availability Statement:** The data underlying the results presented in the study are within the paper and its Supporting information files. Additionally, the data for the figures and tables are

## Abstract

Evolutionary computing (EC) is an area of computer sciences and applied mathematics covering heuristic optimization algorithms inspired by evolution in Nature. EC extensively study all the variety of methods which were originally based on the principles of selectionism. As a result, many new algorithms and approaches, significantly more efficient than classical selectionist schemes, were found. This is especially true for some families of special problems. There are strong arguments to believe that EC approaches are quite suitable for modeling and numerical analysis of those methods of synthetic biology and biotechnology that are known as in vitro evolution. Therefore, it is natural to expect that the new algorithms and approaches developed in EC can be effectively applied in experiments on the directed evolution of biological macromolecules. According to the John Holland's Schema theorem, the effective evolutionary search in genetic algorithms (GA) is provided by identifying short schemata of high fitness which in the further search recombine into the larger building blocks (BBs) with higher and higher fitness. The multimodularity of functional biological macromolecules and the preservation of already found modules in the evolutionary search have a clear analogy with the BBs in EC. It seems reasonable to try to transfer and introduce the methods of EC, preserving BBs and essentially accelerating the search, into experiments on in vitro evolution. We extend the key instrument of the Holland's theory, the Royal Roads fitness function, to problems of the in vitro evolution (Biological Royal Staircase, BioRS, functions). The specific version of BioRS developed in this publication arises from the realities of experimental evolutionary search for (DNA-) RNA-devices (aptazymes). Our numerical tests showed that for problems with the BioRS functions, simple heuristic algorithms, which turned out to be very effective for preserving BBs in GA, can be very effective in in vitro evolution approaches. We are convinced that such algorithms can be implemented in modern methods of in vitro evolution to achieve significant savings in time and resources and a significant increase in the efficiency of evolutionary search.

contained in the Zenodo repository at https://
zenodo.org/record/5704168#.YZOAcflBzik
(Alexander Spirov, Ekaterina Myasnikova. (2021).
"Heuristic algorithms in Evolutionary Computations
and modular organization of biological
macromolecules: applications to in vitro evolution.

**Funding:** The research was supported by the
Russian Science Foundation (grant 17-18-01536).
The funders had no role in study design, data
collection and analysis, decision to publish, or
preparation of the manuscript.

**Competing interests:** The authors have declared
that no competing interests exist.

# 1 Introduction

Prospects for new applications in those areas of computer science and applied mathematics
that encompass the Nature-inspired heuristic algorithms, attract nowadays considerable inter-
est. In particular, such now thriving vast area, collectively known as evolutionary computing
(EC), was once formed as a result of the transfer of ideas and concepts from evolutionary biol-
ogy [1, 2] and over time has been significantly enriched with the methods of probability theory,
complexity theory and optimization (eg [3]). The most attention is paid to such a canonical
area of EC as genetic algorithms (GA), in which many key generalizations are formulated.

## 1.1 Biological in vitro evolution and genetic algorithms

In recent decades, there is an emerging trend for a reverse transfer of approaches from EC to
the systems and synthetic biology [4–15]. Moreover, GAs are proposed to be used as a basis for
mathematical and computer modeling of experiments on directed evolution of biological mac-
romolecules (evolution *in vitro*, SELEX [14, 15, 16–28]. Here we use close in meaning terms
SELEX and directed evolution to generally define the whole area of the evolution of biological
macromolecules in test tube. Directed evolution (artificial evolution of biological macromole-
cules) is a laboratory process which is used to create biological molecules with desired traits by
means of iterative diversification and selection cycles. These manipulations involve the
increase of genetic diversity, sequence evaluation, and selection for required functions. Typi-
cally, experiments are held with large molecule populations.

In EC, the idea of schemata (the Schema theorem [1, 19]) and the theory of building blocks
(BB), laid down by the works of John Holland and colleagues, formulated the principles of the
effectiveness of evolutionary search [20–22]. GAs proved to be effective for those problems of
evolutionary search for which the genotype to phenotype mapping is correctly specified.
Namely, from the very beginning of the evolutionary search from scratch, various short redun-
dant sequences, "schemata", appear in genomes, contributing to the fitness of the genome
(these are lower-order schemata). Schemata are relatively short subsequences of interest within
longer sequences. Genome recombination combines such schemata into higher-order blocks
with greater fitness contributions, called BBs. In turn, BBs can then be combined by recombi-
nation into blocks of an even higher order.

## 1.2 Building blocks in biology and genetic algorithms

In GA, schemata and BBs have clear analogs in the evolutionary biology of macromolecules
(nucleic acids and proteins): numerous hierarchical evolutionary conservative functional
motifs, modules, and domains in biomolecules [23–25]. Evolutionary biologists believe that the
evolution of macromolecules proceeds by the recombination of individual motifs, modules,
and domains into BBs of a higher order. This convergence of low-order domains into compos-
ite higher-order domains/modules can provide a significant increase in fitness due to the
newly acquired characteristics of such new recombinant molecules. This is very reminiscent of
theoretical generalizations of the effectiveness mechanisms in the GA evolutionary search.

As in GA, evolution by assembling low-level schemata into blocks of ever higher levels and
higher fitness through recombination implies special mechanisms for protecting already
found blocks from subsequent destruction (decompositions) by recombinations (crossover) in
the ongoing evolution [16, 17, 26–28].

It is noteworthy that longstanding searches for such EC algorithms, protecting the already
found BB from destruction by crossover and mutations, led to the discovery of a considerable
number of both universal and rather specialized mutation algorithms not destroying BBs [29–
34]. On the other hand, in experimental techniques of *in vitro* evolution, the goal of finding

methods that preserve BBs has never been set (as far as we know). Moreover, the importance of saving time and resources in very costly modern procedures for directed evolution of macromolecules is obvious (and does not particularly need additional detailed argumentation).

All the above considerations bring us to the conclusion that finding of new heuristic procedures increasing the search efficiency by orders of magnitude are of fundamental importance.

Among the ever-expanding variety of methods and approaches for directed evolution of biomolecules, two relatively old experimental areas are closest to the theory and practice of GA. These are SELEX proposed in 1990 [35] and DNA-shuffling proposed in 1994 [36, 37].

The SELEX traits that are important for the following are given in the text-box.

The SELEX-based methods were implemented for the evolutionary search for multivalent (chimeric) aptamers [50–53]. In such approaches, a functional RNA molecule, already successfully selected for one functional domain (for example, aptamer), or even for two domains (aptamer and connector / communication domain), is further selected for the next (second or third) domain (for example, ribozyme) so that the functions of the already found domains are preserved [54–59]. To such a fully functional triplet of aptamer-connector-ribozyme domains (generally called aptazyme), it was possible to add one more aptamer with its own communication block. The result is a five-domain allosteric ribozyme, whose enzymatic function is controlled by its cooperative binding of two different effectors by its aptamers [59].

Besides the sequential search of domains, successful realizations of their simultaneous (dual-selection) evolutionary search are also published. So, Landweber and Pokrovskaya managed to isolate functional RNA with two different ribozyme activities (ligase and self-cleavage activity) from a random pool [60]. As a result of several SELEX rounds, two functional modules, a sensor and a connector (regulatory module) of an RNA device, were simultaneously acquired de novo [61].

The remarkable similarity between GA techniques and SELEX (and more broadly, *in vitro* evolution) is primarily clear from the similarity of procedures. In both cases, the process starts from the initial population, however, the properties of populations may differ significantly (e.g. [61]). It is noteworthy that the basic procedures of the evolutionary search for the classical SELEX aptamers are quite comparable to those in the standard GA [18, 62] (See Text box).

Of particular interest are cases of *in vitro* evolution aimed at finding functional molecules with several mutually interacting functional domains at once (as, for example, in aptasensors and aptazymes). These are the cases when borrowing algorithms from GA for implementation in the artificial evolution of biomolecules is the most promising. This borrowing can be based on the theory of schemata and BBs, where exist efficient algorithms preserving the already found BBs (cf. [16, 63]).

Therefore, new approaches require new theories for finding the desired sequences when they are not present in the initial (and relatively small) populations.

## 1.3 Biomolecular royal staircase functions

What is most impressive is not the striking similarity between GA procedures and the evolution of biomolecules, the really most impressive thing is that the evolutionary search for functional macromolecules in experiment obeys the same laws as the evolutionary search in GA. It was this remarkable similarity that inspired us to this study. The idea of schemata and BBs, laid in the foundations of the GA efficiency concepts, needs to be generalized and extended to the field of macromolecule evolution.

With this aim we defined our *Biomolecular RS* (BioRS) functions so that BBs were assigned the same key basic traits as the domains of artificial macromolecules. Then numerical tests

## Box. SELEX vs genetic algorithms

The aim of SELEX is to find a DNA molecule which is highly specific to a target molecule. This target molecule is bonded to a surface, then washed with the population to only bind with the most specific DNA molecules. Then the solution with unbound molecules is washed out, while those bound to the target are amplified by polymerase chain reaction (PCR). The cycle is repeated many times at gradually increasing temperature, i.e., at increasing the binding energy. Finally, the molecules with the highest affinity to the protein are bound, amplified, and detected in the experiment. Evolutionary search uses an error-prone PCR, in which PCR is deliberately disrupted by chemical and other manipulation, with the error rate as high as $10^{-2}$ [38]. More generally, in directed evolution, this may be combined with the gene shuffling to induce recombinations between DNA strands. Directed evolution, as well as the noisy SELEX, mainly differs from typical evolutionary algorithms in the relatively short runs, and in the huge populations used ($\sim 10^9$ up to $10^{15}$) [39–41].

SELEX is naturally interpreted as a standard GA with a point mutation operator (without crossover), while another well-known method of directed evolution, DNA-shuffling, resembles more exotic versions of GAs with multiple parents [42, 43] and homologous recombination operators [8]. Over the past couple of decades, the SELEX modernizations were directed towards powerful and versatile techniques with great prospects in synthetic biology and biotechnology [44–48]. Modern successors of the classic SELEX, vesicular and microdroplet techniques, are the methodologies in which the implementation of GA operators is most promising. Using these experimental biological approaches as an example, we outline the prospects for the analogues and interrelations between the GA computational test and evolution of biomolecules in the context of the evolutionary study of functional multidomain macromolecules.

However, SELEX demonstrates such search efficiency only for problems aimed at searching for aptamers. The capabilities of the SELEX selection of macromolecules for other properties is extremely limited. More modern approaches based on experimental manipulations with individual macromolecules in artificial cells or in micro-vesicles or micro-drops [44–49] are able to select more and more effective (deoxy-) ribozymes. It is critical that in these modern approaches it is practically impossible to use the main advantage of SELEX: selection in populations of as many as $10^{15} - 10^{16}$ sequences. Population sizes there are several orders of magnitude smaller [45–47]. Large population size is exactly what was initially declared as one of the main and impressive advantages of SELEX, which, as was hoped, would allow to find almost any desired activity in such huge initial populations, so that all that remained was to further enhance the found desired property with subsequent few rounds of mutations and selection [40, 41, 50].

with such functions will be suitable for modeling both existing and putative methods of artificial evolution of biomolecules in biotechnology.

The concept of BBs in ECs has once inspired computer scientists to introduce and develop the Royal Roads/Staircase (RR/RS) fitness functions [1, 19–22, 64]. These functions were used to develop new efficient algorithms in comparison with the standard EC techniques. Evolutionary search with the RR fitness function is organized so that the fitness value grows stepwise only after a whole block is found. In the RS modification, the search of blocks is performed in a strict order, one after another. (For details, see S1 File "Royal Road functions").

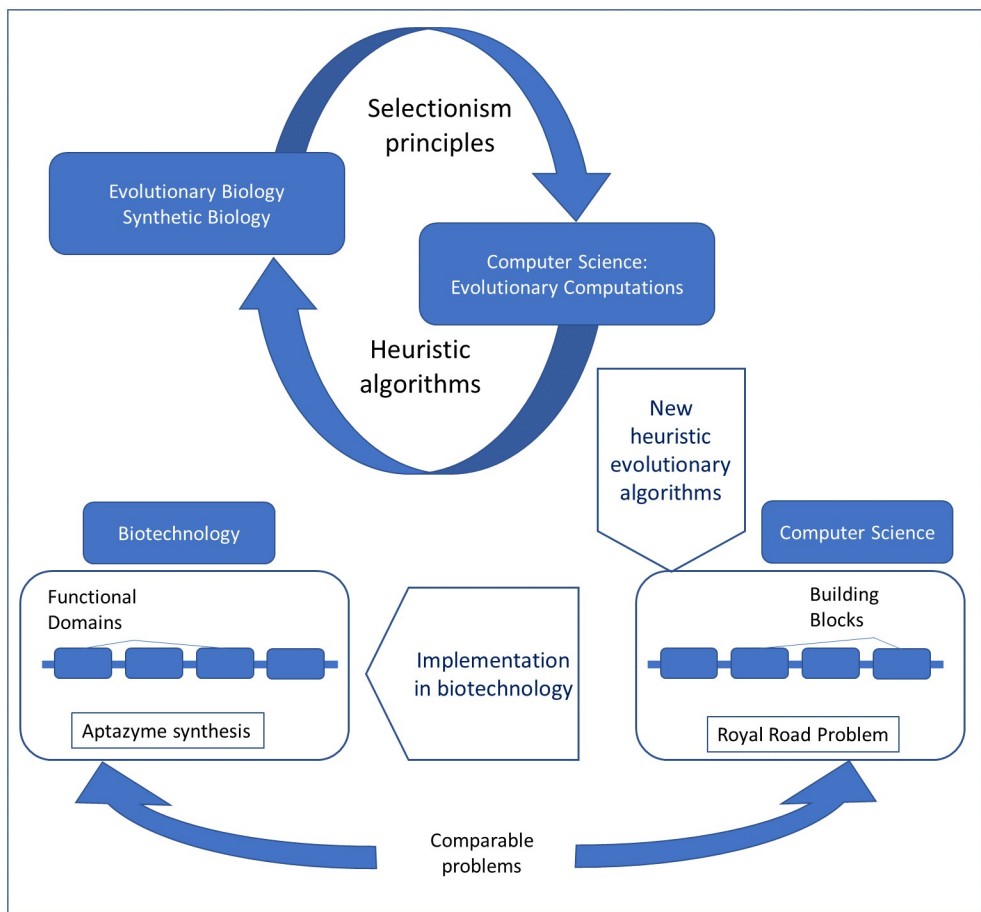

**Fig 1. Heuristic algorithms designed in EC have great prospects for the transfer to synthetic biology and biotechnologies.** Namely, the impressive similarity between the problem of the synthesis of multidomain (multifunctional) macromolecules from scratch and the RR problem suggests that it is promising to transfer and implement these heuristics into new experimental techniques.

The general search problem for full-featured multidomain RNA devices has clear analogies with the RR/RS problem, therefore, those algorithms from among GAs, which correctly simulate the synthetic biology techniques and are much more efficient, may turn out to be very promising for the practical applications of directed evolution. In this article, using our extension of RS functions for the problems of macromolecule directed evolution (BioRS functions) as an example, we test the most effective of the simple heuristic algorithms in order to propose ways of their implementation into the practice of synthetic biology. An idea underlying our approach is illustrated in Fig 1.

It is natural to consider the SELEX search from scratch for multiple domains (aptamers) as the main control in our series of the BioRS numerical experiments. By now, a series of papers on the SELEX theory and, more broadly, *in vitro* evolution has been published [18, 62, 65–81]. However, in fact they all consider the search for a single domain, usually from scratch, and usually under the assumption that the initial population already contains single copies with the desired activity, albeit low but detectable. Hence, the evolutionary search is typically just aimed at gradual improvement in fitness and further multiplication of the best copies found. This is the main reason why it is necessary to borrow and develop new heuristic algorithms designed in GA for solving problems of stepwise search for multidomain molecules.

Our goal in this article is to continue in-depth studies of the prospects for the transfer of ideas from E to synthetic biology and biotechnology, demonstrating these prospects with compelling examples. The two major tasks in this context are to design experiments *in silico* for numerical testing the effectiveness of known from EC heuristic algorithms (1), and show that there exist algorithms significantly more effective in *in silico* tests than those traditionally applied to model *in vitro* evolution (2). We believe that implementation of our results will be useful in the further development of new effective, less expensive and less resource consuming biotechnological methods.

## 2 Results

In this section, we introduce our extensions of the RR fitness functions adopted from the GA theory, with special focus on their RS modification (See Methods). These functions, which we call BioRS, will be generalized to the case of *in vitro* evolutionary modeling. Then we present the results of testing several heuristic algorithms with BioRS functions. Our goal is to find numerically such evolutionary algorithms which appear to be more effective in modelling standard experimental *in vitro* techniques.

### 2.1 BioRS fitness functions for directed evolution *in silico*

In this research we introduce BioRS, our extensions of the RS functions bringing them closer in properties to the structure of biological macromolecules. These extensions retain the basic features of the «progenitor» RR/RS functions but acquire new characteristics of the BBs as biomolecular domains.

When defining and implementing the RR/RS functions within the framework of the *in vitro* evolution theory, we first of all assume that the sequences are not binary, but *quaternary*. Moreover, due to the non-binary nature of biological macromolecules, the concept of a schema should be also extended. The schemata are typically *redundant*. A binary schema is a set of bit strings that can be described by a template made up of three types of elements 0, 1 and * (a wild card). In the case of nucleic acids with a four-digit code, this set expands to 15 elements (four single-valued elements, A, T, G, C; six pairs of equally (or maybe unequally) possible elements; four triples of (un)equally possible elements and an * element). Such molecular biological schemata are well known to biologists. Due to their understandable complexity, graphic means of their presentation were developed, known as logos.

Besides, our functions can consist of end-to-end blocks (like RR1 functions used in ECs) or can include *spacers between domains* (analogously to RR3 functions). See S1 File «Royal Road functions» for details.

Finally, a BB in the concept of BioRS functions *is not a unique sequence, but a set of closely related sequences* corresponding to the same consensus (calculated order of most frequent nucleotides found at each position in a sequence alignment) and with close, high or high enough, fitness values. This is an essential generalization. In the case of canonical RR functions in evolutionary search, the fitness grows stepwise by a given value only after a single sequence (new block) is found, i.e., is defined as proportional to a number of already found blocks. In the case of BioRS, the next fitness level is awarded to each out of a whole set of sequences, with close enough fitness values. In EC, the period of search for a next, *n*th, block is referred to as an *n*th *epoch*, and we will also use this term. The fitness function parameters are defined so that distances between adjacent fitness levels (between epochs) are significantly higher than distances within a given level (within an epoch). Thus, the steps in the fitness landscape become smoother. We will refer to this search property as "*graduality*".

In our approach we take into account the inequal occurrence of different nucleotides. We assume that the affinity to the more preferable variant of consensus is awarded the higher fitness, which gives us the advantage that the search reveals domains of the higher binding ability. The implementation of the BioRS function is described in Methods.

We take aptamer batteries designed and researched by Babiskin & Smolke [82] as an inspiring example for the implementation of a specific version of BioRS, as described and illustrated below. This example is rather didactic; however, a number of authors investigate aptamer batteries as having good prospects for use, primarily in medicine [83, 84]. This example is well suited to expand the definition of the RR/RS functions to the field of *in vitro* evolution. Moreover, the natural further expansion of our BioRS would be batteries not of the same, but different aptamers (specific to different ligands).

To elucidate our approach, we take a small-sized classic and well-studied ATP-binding aptamer (binds adenosine and its derivatives). This artificial aptamer was independently isolated several times by the SELEX technique from a random pool and was first found by Sassanfar and Szostak [85, 86]. In literature, it is often referred to as the *Sassanfar–Szostak motif*. Set of sequences similar to the consensus is well studied [86–88]. This is illustrated by the consensus in Fig 2B and 2C.

It should be noted that sequences of the modules/domains of biological macromolecules are redundant by definition. Moreover, the redundancy of the main domains of RNA devices has a complex nature (See Fig 2). We intend to implement at least the basic components of this redundancy in our BioRS. Firstly, both sensors and effectors have a (redundant) motif in the functional part of the module. In the case of an aptamer, this is its "core" part, which provides specific recognition and binding of the ligand (See Fig 2B and 2C). This core is typically highly conservative. Secondly, both sensors and effectors necessarily have structural elements (Fig 1B). These are stems and loops that form the aptamer (or ribozyme) scaffold. They are less conservative in sequence. In an aptamer structure, conservative functional components are interspersed with less conservative structural elements (Fig 2B and 2C). We implement all this in a simplified form to use as a specific example of the evolutionary search with BioRS.

Preliminary runs showed that the original version of BioRS (based on the Sassanfar-Szostak motif) is still too complex for evolutionary tests and is beyond the capabilities of modern desktops and workstations. Therefore, we had to somewhat simplify the task. In fact, we are only searching for the sequence of the most important functional part of the aptamer (responsible for recognition and specific binding of the ligand). Whereas we consider the structural hairpins already given. As a result, the domain-aptamer of our BioRS function is reduced to 26 bases, of which we search 4 bases as uniquely defined, and 10 as two-base varying, the rest of the bases are arbitrary. This corresponds to the finding of one sequence from 262 144. The specific logo of the consensus we are looking for is presented in Fig 2D.

In our numerical experiments it is assumed sufficient to find any sequence from a whole family of those close enough to the consensus to increase the BioRS value up to the next fitness level. In other words, the search is transferred to the next epoch and is aimed to find the next domain.

The corresponding evolutionary search scheme is shown in molecular biological details in Fig 3B. Thus, the evolutionary search in the case of our BioRS function is very similar to that for the prototype RS3 (Fig 3C; See S1 File).

Note that spacers for experimental manipulations with molecules are located between the stretches containing the already found domains; the spacers are assumed to have no function.

As one can see, the evolutionary search with the BioRS function with spacers can be interpreted in terms of the modern biotechnology and synthetic biology. At the same time, it is formalized in such a way that we can easily implement it computationally by means of GA as

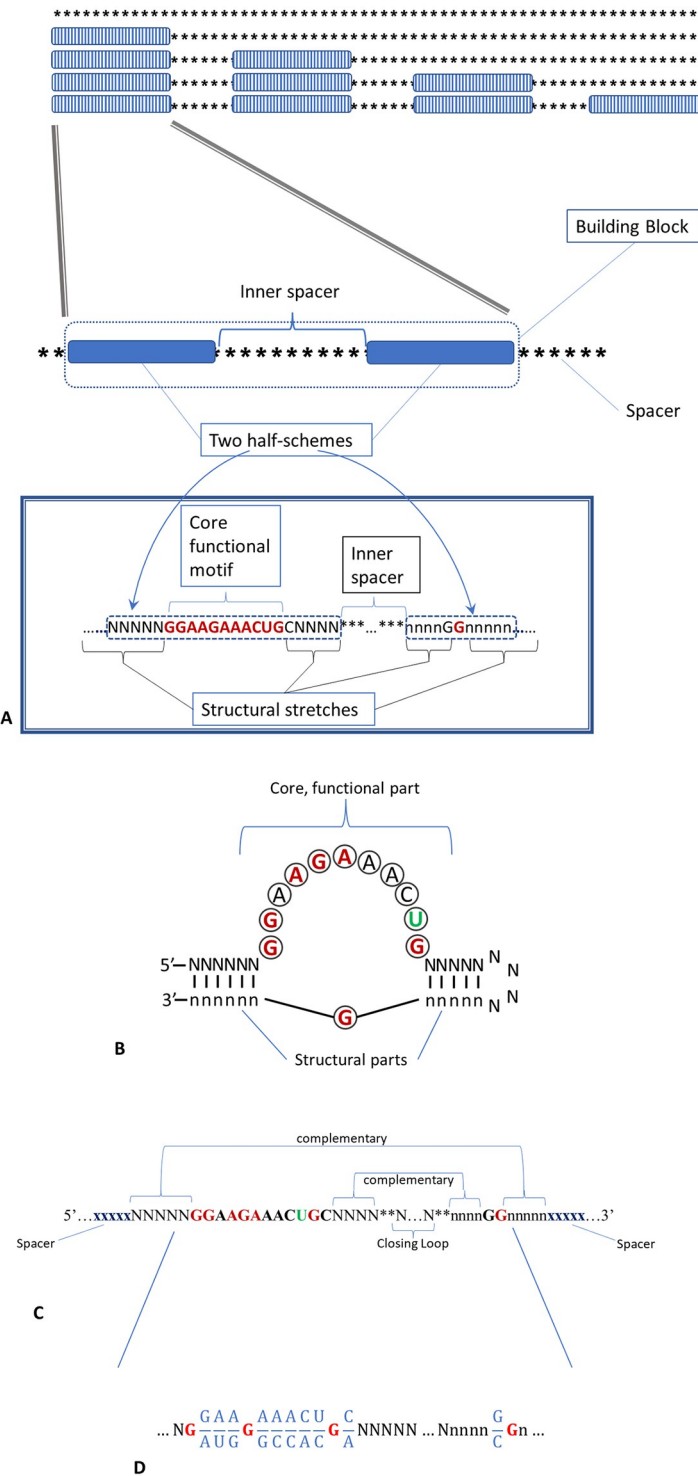

**Fig 2. The specific implementation of the BioRS function in the molecular biological aspect.** Aptamer is presented as a composite BB of two half-schemes, which is representative for the organization of aptamers. The inset shows the internal structure of each half-scheme of the Sassanfar–Szostak motif [85, 86]. See text for details. All the domains are assumed to be identical. (B-C) Structural details of conservative functional regions of an aptamer using Sassanfar–Szostak motif as an example: (B) 2D structure and (C) nucleotide sequence. Positions in the recognition bulge are colored according to whether they are invariant (red), two-base varying (green), or non-conserved (black). Here . . .xxxxx corresponds to the halves of the insulating spacer sequences, NNNNN & nnnnn are the complementary halves of the sequences of the first and second stem, ** . . .** is arbitrary and corresponds to the closing loop sequence.

(D) 26 nucleotide core part of the Sassanfar-Szostak motif implemented in our BioRS as a consensus. Four G-nucleotides of this consensus are single-valued (red), 10 loci are defined up to nucleotide pairs (blue), while the rest are assumed arbitrary (N or n, in black).

follows (Fig 3C). An initial sequence of the predetermined length is chosen arbitrary (a random nucleic acid molecule) and does not have any motif fitting the consensus. The initial goal of the evolutionary search is to find the first aptamer sequence, the far left. After the first sequence is found, the sequence of the second aptamer, next to the first, becomes the goal of the evolutionary search. Upon finding the sequence of the 2nd aptamer the fitness of the molecule increases, and the evolutionary search switches to the 3rd aptamer, located downstream the already found ones. In the described formulation of the problem, we consider the spacers to be given (e.g., by random sequences). So, we are looking for the domains in the strictly specified positions and in a strict search order (from left to right).

Here we are implementing a sequential rather than parallel evolutionary search for domains, as our ultimate goal is the practical applicability of these studies. In an experiment, especially in the search for domains of RNA devices, the search for the next domain depends on the structure and position of those already found.

It is known that, in the general case, stitching into a complex of two or more domains found earlier and functional under the conditions they were found does not necessarily give a new functional complex. For example, stitching of two aptamers working separately results in their loss of function due to mutual interference [51, 52, 89]. On the contrary, if we are looking for the entire multi-domain fully functional complex (for example, aptazyme) from scratch, identifying successively domain by domain, then the situation changes dramatically. Each new domain is searched for under the obvious condition that it will not interfere with the functioning of those already found. Therefore, a new domain will be selected from those alternatives that satisfy this condition. Hence searching for a functional RNA device from scratch is better than stitching together separately found domains into a composite complex.

**2.1.1 Functional domains of biomolecules are generally not attainable by a gradual search in the initial pool.** We focus here on a very important characteristic of functional domains of biomolecules, which relates them to BBs in the RR/RS functions. In the directed evolution of macromolecules, it is believed that a sufficiently large initial library will necessarily contain a sequence with a desired function, albeit maybe very weak but sufficient for identification (for example, a function of a particular aptamer or ribozyme) [40, 41, 90]. Then the further artificial evolution is reduced to a gradual search for more and more powerful versions of these functional domains.

In fact, in the general case, this is not true, and typically the initial library will not contain suitable starting sequences for the following gradual search. On the contrary a search is performed through zillions of mutant sequences before at least one such sequence with the required function of a level sufficient for subsequent gradual selection is randomly formed by means of mutations / recombinations. Due to the fact that typical aptamers and (deoxy) ribozymes are up to several hundred nucleotides (150–300 np) in size, the finding of consensus by brute force will take up to $10^{90} - 10^{180}$ steps, we have all grounds to believe that it is reasonable to begin the gradual search only among sequences relatively close to the consensus that can not be present in the initial population.

Therefore, in the proposed version of BioRS, we proceed from the concept of consensus. So, the simulated evolutionary search aims to find a relatively small and compact group of sequences that fit the same consensus. Only these sequences are awarded positive fitness values while more distant ones are ignored.

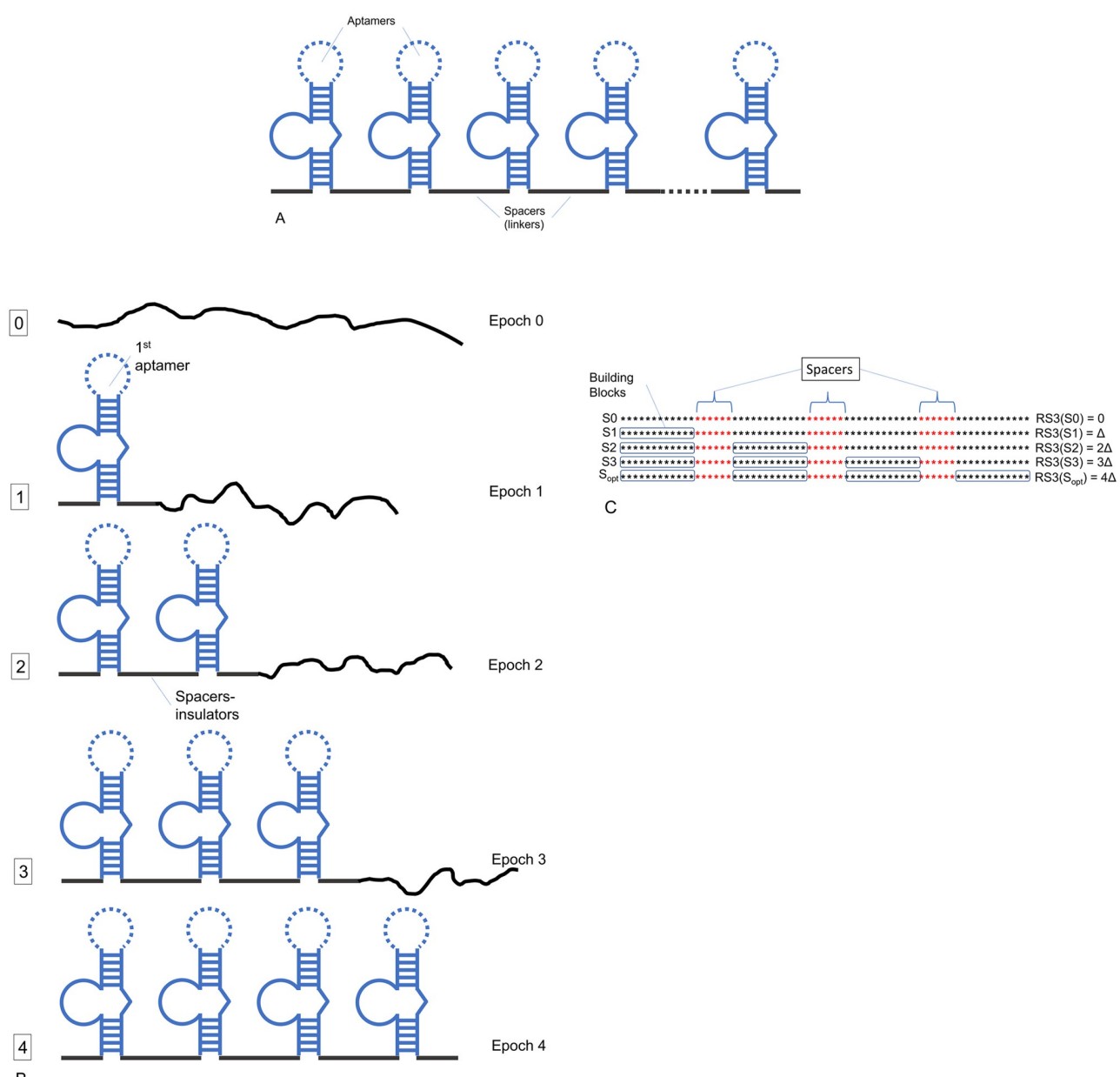

**Fig 3. Molecular biological illustration of the idea of evolutionary search with BioRS.** (A) General idea of the battery of (identical) modules as it is formulated by Babiskin & Smolke [82]. (B) Our scheme of sequential search from scratch of a tetrameric aptamer. (C) Search scheme for a classic RS function with 4 BBs (above). Domains (BBs) separated by insulating spacers of predetermined length are searched sequentially from left to right. Delta is an increment of fitness level for each correctly found aptamer (BB). See text and legend to Fig 2 for the detailed description of an aptamer structure.

**2.1.2 Biologically substantiated extension of BioRS function.** The simplest version of BioRS, based on the concept of consensus, is defined in such a way that all sequences fit to the consensus are assigned certain fitness added to the standard step RS function only taking discrete values characterizing the number of already found domains. If we assume that different consensus variants unequally contribute into the fitness function, our BioRS will take the gradual form, i.e., it can have intermediate fitness values. The detailed description of this function

is given in Methods ("onsensus based BioRS function") using an example of consensus presented in the top of Fig 4E.

This BioRS, referred to as a *simple consensus-based*, is in a natural way generalized to describe more complex structures and interactions, characteristic to macromolecule domains, such as possibility of alternative versions of an aptamer to a certain ligand and interference between the domains. Finally, we summarize all these extensions in the most realistic version of BioRS which we call *competitive*. It approaches in detail and complexity to real-life problems, and its further development and research should have significant prospects when applied to modern approaches of directed evolution.

*2.1.2.1 Consensus-based BioRS function with alternative domain structures.* Functional domains of macromolecules may typically have more than one possible structure (with different consensus and different secondary–tertiary structures). Domains are not just slightly different in sequence. There exist several qualitatively different domain structures with similar or even almost identical functions. For example, several qualitatively different structures of ATP-binding RNA aptamers have been found and characterized [85, 88, 91–93]. Specific versions of alternative ATP-binding RNA aptamers, which we were inspired with, are shown in Fig 4A and 4B. We found it necessary to implement this property in our BioRS in order to bring it closer to real-life problems.

An example is shown in Fig 4E (two lower panels). This approach is a generalization of the simple consensus-based BioRS to two possible (alternative) consensuses, Structure-I & II, that are for simplicity assumed to be of the same level of fitness. We also assume that the fitness of a new domain is the higher, the closer it is to one of the consensuses, while the affinity to the alternative consensus is not taken into account (see Methods for details). With such a simple formulation, the evolutionary search for the current domain ultimately leads to finding one of the alternative consensuses. The fitness function in this case takes on a maximum value for two different sequences of the entire domain, i.e., it has two maxima.

*2.1.2.2 BioRS functions with domain interference.* An important property of artificial selection for multidomain functional RNAs (for example, pairs of aptamers to different ligands [51, 52]) is the possible interference in the search for a new functional domain with those previously found. Namely, already found functional domains may not allow finding a new functional domain. In less severe cases, finding of a new functional domain can reduce the fitness of those already found. For example, Burke and Willis [51] fused a pair of aptamers into a chimeric molecule and this led to a significant decrease in the affinity of each of these two aptamers. A chimera from a pair of aptamers in Wu and Curran [52] also demonstrated the competitiveness between the aptamer functions. In this version of BioRS, the fitness is awarded as described for the consensus based BioRS but taking into account the domain competitiveness. From the point of view of molecular biology, this means that each new domain can interfere with those already found and thereby interfere with their normal functioning. This is primarily due to the fact that domains block normal folding of each other (cf [51, 52, 89]).

Accordingly, when searching for a new domain, it is necessary to reconfigure the entire ensemble of already found domains (through the evolutionary search for their new / alternative consensuses). In this case, a gradual search in the general case will not be possible, because some of the already found functional domains prevents finding a new domain (in a given range of positions and within a given range of its length).

In general, a domain can have several dramatically differing consensuses. Moreover, a given functional organization of a certain domain typically corresponds not to a unique but to a family of consensuses, mainly due to several structural elements, size and sequence details of which, are of little importance for the domain functioning. However, these are characteristics that can be essential for interaction with neighboring domains. This is why a new domain can

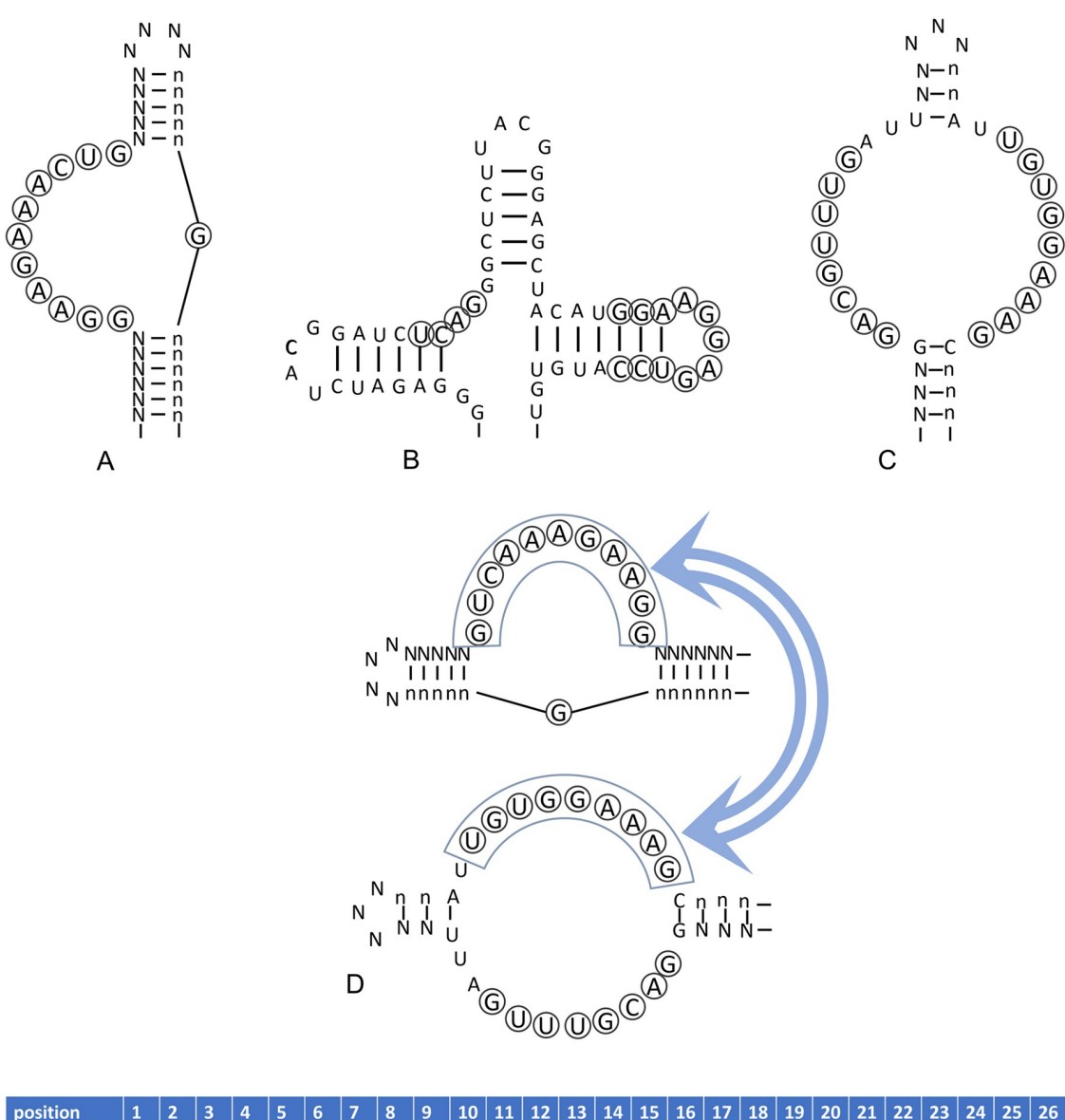

**Fig 4. General idea of alternative structures of aptamers binding the same ligand and its specific implementation scheme for the BioRS function.** (A-) Three structures of aptamers binding to ATP that differ in their sequence and secondary–tertiary structure. The first structure (structure-I) (Sassanfar-Szostak motif) we will name as a "rod" (A). Second structure (structure-II) will be referred to as a "cloverleaf" (B) [93]. () One more version of an aptamer binding ATP [92]. (D) Consensus sequences of (A) and (C), critical to the recognition and binding of a ligand, in spite of their general similarity, differ to such extent that it is infeasible to transit from one to the other by a series of mutations not losing the aptamer functionality (sequences similar in both versions are framed and marked by double-headed arrows). (E) Single- and double-valued loci in our simplified version of the aptamer with alternative structures. Single-valued nucleotides, necessarily present in both structures, are given in red, those present in one of the alternative structures in black. The other positions are arbitrary. See text for details.

be only detected for some specific versions of sequences of the other domains, while the currently found best versions, despite their high fitness, may be dead-end evolutionary solutions not allowing for the further search. Hence, it may happen that the evolutionary search will be only successful due to suboptimal versions of already found domains.

*2.1.2.3 Competitive BioRS function.* Finally, we further extend the concept of our BioRS functions. This extension, which encompasses alternative consensus and domain intervention, will be referred to as *competitive* BioRS for brevity. The search scheme for this particular version of the extended BioRS is shown in Fig 5. As an illustration, two versions of our aptamer are presented: Structure-I is "rod" and Structure-II is "cloverleaf", where the cloverleaf is significantly wider than the rod (see Fig 4A). We are using a simple "didactic" version of the mutual influence of neighboring domains. Namely, steric interactions between a pair of neighboring domains are such that a next domain can be only found if the structure of the preceding, already found, domain has a rod structure. If the previous domain is a cloverleaf, then further search is impossible (Fig 5B and 5D). At the same time, the new desired domain itself can have any of two structures (as in Fig 5A–5C).

Note here that this is just a didactic example, and it is impossible to obtain a rod or clover from specific sequences in Fig 5D, since this requires much more defined positions and longer sequences. However, the implementation of such a task in full it is too complex for real workstations, so we use a very simplified version.

In terms of fitness functions, a new domain can only have a nonzero contribution to the fitness if the previous, already found domain has a sequence fit to the first consensus ("rod") (Fig 5C, 5E and 5F). Otherwise, if it fits to the "cloverleaf" consensus, the further evolutionary search is only possible by annihilating the fitness of the already detected domain by mutation and finding an alternative consensus for it. A specific example of the competitive function for an aptamer with alternative consensuses (Fig 4E) can be implemented as a BioRS with the alternative structures and domain interference, which is described in Methods ("BioRS functions with domain interference").

In this article, we will test two versions of the BioRS function numerically for the search of the simplified Sassanfar-Szostak motif battery. First, the simple consensus-based BioRS, which corresponds to the situation where only one domain structure is allowed (no alternative) and it is possible to find a next domain not interfering with those already detected in their folding and functioning. This version of our function is interpreted as a formalization of classical experiments to find the next domain if other domains have already been found in the evolutionary design of aptazymes [54–59].

Second, it is competitive BioRS describing a more complex, but more general task, which may well be encountered in real life. This is the situation where the search and finding of each new domain presuppose changes of the sequences of already found ones in order to eliminate their antagonistic mutual influence. It may happen when there does not exist such a structure of a new domain that does not interfere with the already found functional domains. Our results demonstrate that these two families of BioRS evolutionary search problems turn out to be quite different.

**2.1.4 Variation of the domain position.** In spite of the similarity of autonomous functional domains/blocks of macromolecules, they critically differ from the RR/RS building blocks in their principles of genotype-phenotype mapping. This provides new ways to accelerate the evolutionary search for BioRS, in particular, and *in vitro* evolution, in general, as will be shown in this article.

In the theory of schemata and BBs, the GA mapping is typically organized in such a way that the phenotype characteristics are determined by the uniquely given position of the symbol in the sequence ("chromosome"). Only a strictly defined position of a schema and BB on the

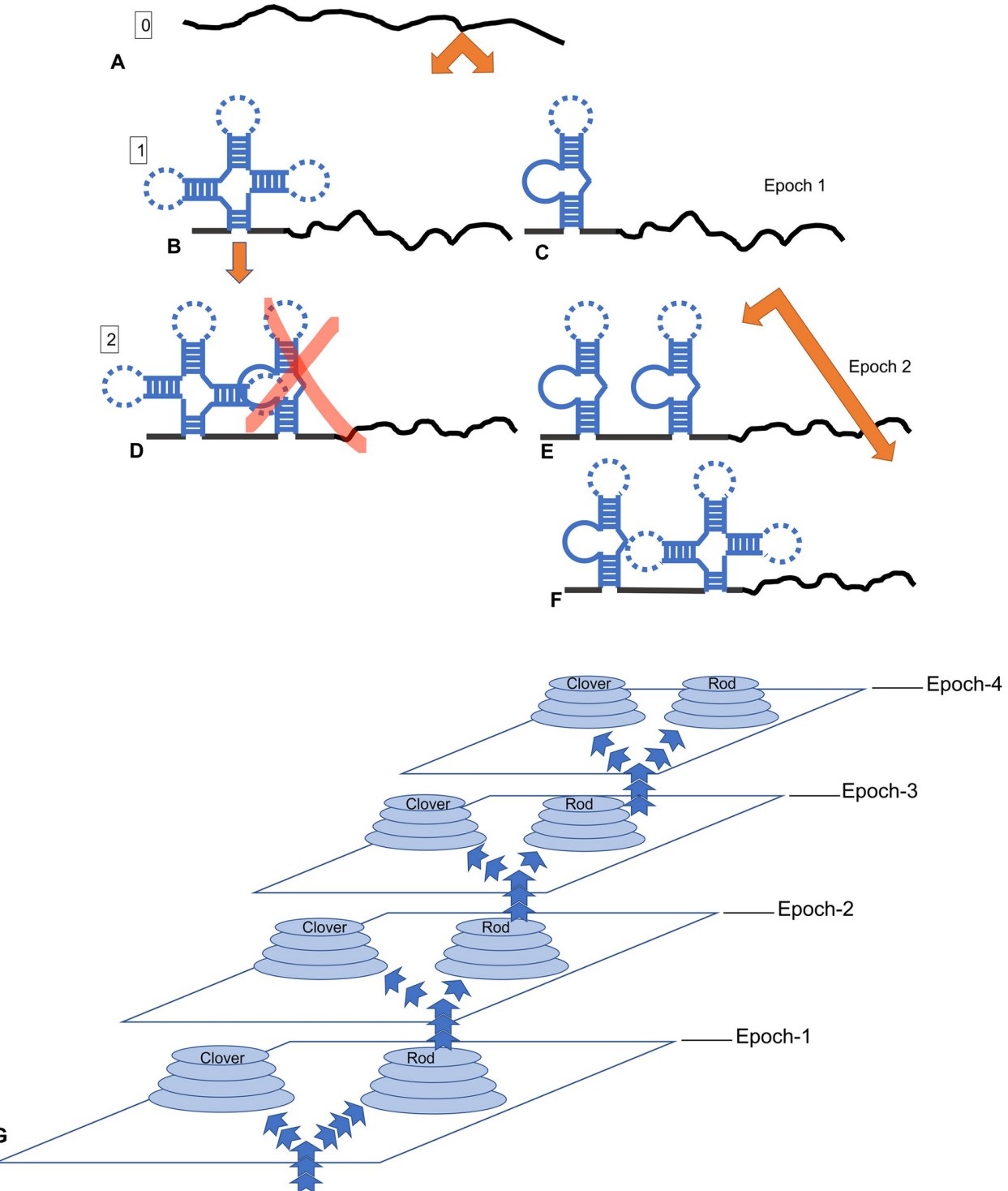

**Fig 5. Extended (competitive) version of BioRS, implementing alternative aptamer structures and interference.** (A-F) Evolutionary search scheme for the competitive BioRS. (A) An initial arbitrary sequence of a given length with zero fitness. (B-C) The first functional domain (aptamer) is identified by an evolutionary search. It may have one of two alternative structures: cloverleaf or rod (see Fig 4A). (D) If the cloverleaf structure is found in a certain position, this makes it impossible to find the second functional domain due to steric interactions of two neighboring domains. (E-F) In the case of a rod structure of the already found domain, it is possible to find the second functional domain in one of two alternative structures. This is only case (E) which can evolve further. (G) Schematic of the competitive BioRS illustrating its multimodality (bimodality within each epoch). See text for details.

"chromosome" determines the level of fitness, any shift in position leads to a loss of block and decrease in fitness.

In molecular biology, on the other hand, functional domains are often separated by non-functional (structural) spacers of significantly varying length, so that domains can be shifted along the macromolecule without losing the level of fitness. This is explained by the fact that the phenotype of a macromolecule is its 3D folding, which is largely determined by the molecule sequence. The 3D structure of domains is defined autonomously, so they can be moved along the sequence, swapped with each other and reoriented, all without significant loss of fitness. These differences in the principles of genotype-phenotype mapping can significantly affect the effectiveness of evolutionary search. Thus, in molecular biological experiments, a domain is searched for within a certain window (the range of its positions) [85, 88, 91]. Therefore, the variability in the domain position and internal spacers of BioRS function critically distinguishes such biological functions from its RS counterpart in GA.

These molecular biology-inspired extra features of our BioRS make a significant contribution to the speed and efficiency of the search. It is essential that an aptamer can be located anywhere within a whole window, as illustrated in Fig 6. In other words, not a strictly specified sequence is searched for, but one of a family of sequences differing in the locus of the domain origin. Similarly, the variability of the length of the internal spacers, such as a closing loop in Fig 2A, further increases the redundancy of the domain sequence of the desired domain and reduces the search time.

It is intuitively clear that the SELEX search for the desired motif with an undefined position on the sequences exceeding it in length should accelerate the search. Simple calculations show that the chance of finding the desired motif in longer sequences is estimated as $4^K \cdot W^N$, where $K$ is the number of defined bits per block/domain, $W$ is the window length, and $N$ is the number of blocks/ domains. In other words, the shorter the block and the longer the window, the less chance.

Our numerical experiments confirm theoretical estimates of the contribution of W to the efficiency of evolutionary search for specific BioRS tests. The results are shown in the (S1 Fig and S2 File). Therefore, in all tests below, we used the search window (specifically W = 220) to speed up the search and make the test problems more realistic.

At the same time, in molecular biology, there are many examples of macromolecules with the rather rigidly defined length of spacers and physical order of the domains. For example, this is critical for organizing the promoter region in bacterial genes, as well as in higher organisms. The principles of mapping for such a functional macromolecule become similar to those for GA. The promoter regions are analyzed and synthesized by *in vitro* evolution, therefore our theoretical foundations of the experiment effectiveness are relevant in this area as well

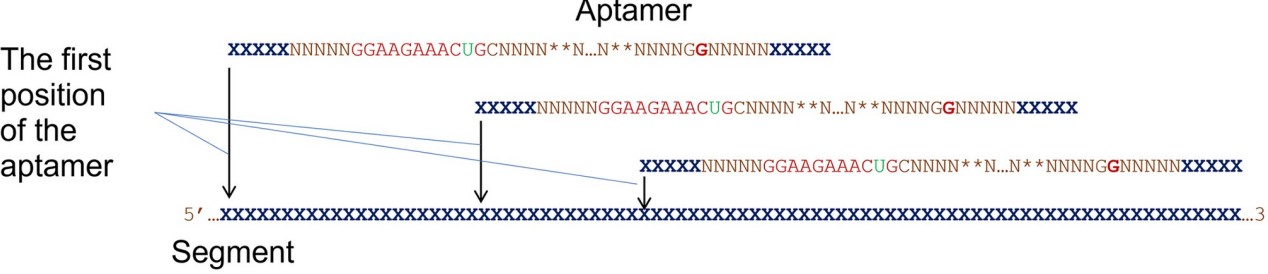

**Fig 6. Idea of the search for a domain/motif within the range from 1 to W positions (in the window) instead of an only predetermined position.** The first position is shifted by w = 1 . . .W nucleotides, where W is not too large (100–200).

[94, 95]. This scheme naturally generates a sequential block search with the BioRS fitness function.

**2.1.5 Biology-inspired techniques accelerating the evolutionary search.** The specificity of *in vitro* evolution techniques, aimed at searching for functional multidomain macromolecules, makes it possible to use experimental approaches of local directed mutagenesis. S2 Fig illustrates the idea. Namely, at each search stage (epoch) just a subsequence of a "molecule", including already found and currently searched domains, is subjected to mutations generating a new candidate sequence. If we are dealing with a non-competitive BioRS, then only that segment of the sequence (far right in the figures) undergoes mutations, where a new domain is currently sought. The sequence segment with previously found domains is protected from being mutated. In the case of competitive BioRS, not only this new segment but also the segment with the previously found domain undergoes mutations.

## 2.2 Standard GA with mutations only: SELEX model

In the numerical analysis of *in vitro* evolution, even relatively simple problems of finding small functional multidomain macromolecules are of the unimaginable computational (combinatorial) complexity. To show this, we will consider the problem of sequential search for four and eight aptamers based on Sassanfar–Szostak motif (as in Figs 2 and 3).

In the previous section, we formulated a didactic version of the BioRS problem that is only based on Sassanfar–Szostak motif (without structural hairpins). To find four motifs, we need to estimate on the order of $4^{32} \approx 18\,446\,744\,073\,709\,600\,000 \approx 2 \times 10^{19}$ sequences. This is twenty quintillion (19 zeros), which is about a million times more than can be estimated by SELEX procedures in one experiment. Below we will show that the use of effective heuristic algorithms, together with the techniques specified by the mapping type (genotype-phenotype) of functional macromolecules, as well as some practices of *in vitro* evolution (for example, local mutagenesis), enables us to reduce this huge number just to about hundreds of thousands estimates of mutant sequences.

In this article, we will use as a control the non-optimized versions of evolutionary search close to typical SELEX procedures for subsequent comparison with GA approaches optimized for the RR/RS problems. For all the simplicity of the implementation of standard GA with mutations only, they still contain a number of parameters and characteristics of procedures that are critical for the effectiveness of evolutionary search. These include, first of all, the choice of the selection scheme and such parameters as the mutation rate and population size. Therefore, we will consider them in more detail below.

Traditionally, the search efficiency (speed) is measured as the number of the fitness function evaluations necessary to achieve the required fitness value. In our case it means that the search is performed until the sequences of all the blocks are detected with the required precision. This is the characteristic which will be used throughout the article to compare the performance of different algorithms in solving BioRS problems.

**2.2.1 Selection schemes/procedures in in vitro evolution.** In EC in general and GA in particular, selection procedures are of great diversity, clearly defined and studied in detail. On the contrary, in SELEX and in *in vitro* evolution, typically, an obvious truncation selection is used with parameters usually not clearly and quantitatively defined, while the role of these parameters is still insufficiently studied both by experimenters and theorists. This is largely due to the specificity of biochemical/ molecular biological methods and imposes its own limitations on the way of modeling and analyzing such experimental approaches. First, although procedurally in the experiments the fraction of the best molecules is taken (with a sufficiently high affinity, if we are talking about aptamers), the (bio) chemical specificity corresponds not

to a strict threshold of affinity in selection, but to the sigmoid binding probability curve [62]. Therefore, using a rigid truncation selection threshold in our models is a simplification of the real situation. Second, this threshold, or more precisely, the parameters of this probability curve [62]), are usually unknown, not explicitly measured or controlled, and vary from experiment to experiment and from cycle to cycle. Detailed description of the variety of selection schemes is given in Methods.

Experimentalists can increase the threshold level from cycle to cycle in order to increase the pressure of artificial selection, but specific quantitative threshold measurements are usually not available. Typically, the number of molecules selected for the next cycle is calculated from the value of the aptamer-target dissociation constant, $K_d$, which is unique for each selection, but its order can be estimated experimentally. For example, for aptamers binding to low molecular weight targets, K_d is assumed to be in the range of tens of μM, and for aptamers to proteins, it is rather in the middle nanomolar range (~10–70 nM). Also, a certain fraction of the molecules goes to the next round of selection due to background binding. Since the proportion of these background molecules is not too large and can be even more significantly reduced by special procedures [96, 97], we do not take them into account in this article.

In order to be as close as possible to the specific biological procedures, we can use typical $K_d$ values from SELEX of ATP & GTP-binding aptamers and compare them with those for aptamers to other small molecules. As a result, the analysis of publications gives us a range of mutation rates from less than a percent to tens of percent. These estimates correspond to those in the classic work of Irvine, Tuerk, and Gold [98] on SELEX modeling, where a realistic fraction of the population, bound to targets, is considered to range between 0.2% and 20%. In the seminal work of 1990 [35] it is stated that less than one-tenth of the RNA from this step was used in the next cycle.

Our preliminary studies have shown that evolutionary search with BioRS functions requires careful analysis of the effectiveness of procedures and approaches. Otherwise, we are faced with a well-known situation in EC practice, when the efficiency of evolutionary search barely exceeds the efficiency of the brute-force search [99]. Keeping in mind the combinatorial complexity of BioRS-like problems, we can face huge waste of resources and very ineffective search in an unoptimized experimental procedures.

**2.2.2 Mutation rates.** The choice of the optimal mutation rate is significantly defined by the specific problem [65, 68]. In practice, it is often assumed that the rate of one mutation per individual per generation taken equal to $1/L$, where $L$ is a chromosome length, is close to optimal [100, 101]. This choice is confirmed by theoretical analysis [102]. Experimental biology (including directed evolution) also has estimates of the optimality of mutagenesis, converging to $1/L$, although extensive and detailed studies of recent decades demonstrate that the rates of point mutations in natural and artificial selection can be significantly higher [103–105]. Note that in problems like our BioRS, when we need to detect not the sequence in the whole but just domains/blocks within it, the optimal average mutation rate is one mutation per the total number of the defined positions.

On the other hand, in SELEX / *in vitro* evolution, experimentalists usually use PCR modifications as a source of predominantly point mutations. In combination with error-prone PCR [106] or hypermutagenic PCR [107], SELEX introduces point mutations that may enhance desired binding at a frequency of ~1–10% per base per PCR reaction.

**2.2.3 Population size and number of cycles.** One of the fundamental limitations of modeling SELEX / *in vitro* evolution is the size of macromolecule populations in experiments on directed evolution that cannot be higher $10^{15} - 10^{16}$. It is this size that largely determines the known effectiveness of the evolutionary search in experiment [41, 90]. On the other hand, numerical experiments on modern PCs and even workstations are also limited by the size of

the simulated macromolecule populations. Even if the molecules are implemented as symbolic sequences (4-letter for nucleic acids or 20-letter for proteins), then numerical tests with populations of $10^7$ are at the limit of computing power and require many days of runs [94, 95].

Note also that the number of cycles in SELEX typically does not exceed 20, while *in silico* tests with the BioRS-like problem usually require many thousands of cycles (cf. [94, 95]).

When starting computational experiments, we must first decide on the choice of a selection scheme. Proceeding from the truncation selection adopted in SELEX, we assume that a reasonable starting point for the design of our *in silico* tests with the BioRS function is the $(\mu, \lambda)$ scheme, which is described in Methods. It is also reasonable to preliminary choose the range of mutation rates from one mutation per sequence per cycle to $1/NK$ mutations (where $N$ is the number of blocks, $K$ is the number of defined positions per block). As a control, we take a test with 8 BioRS blocks (with 6 single-valued and 4 double-valued positions out of 26+220, the search window size W = 220) and at different values of population sizes. This effect is mostly noticeable for short domains. With the increase in the number of defined positions the influence of population size diminishes. The size of the population is limited by computing power but even population 200,000 is very small as compared to SELEX. The results of this test are shown in Table 1. These specific search terms, as well as the search performance for them, will be the starting point for our computational analysis. The search efficiency is more than 140 million fitness evaluations, and the success rate is ≈70% with the number of evaluations being limited to 200 million (Table 1).

We believe that we have reason to look for our minimal aptamer of 26 length on sequences up to 220 bp, because in Zhostak's laboratory several ribozymes once were found by in vitro evolution using initial RNA pools with random parts of molecules up to 220 bp [108]. The same length random sequence has been used, for example, by Ekland and Bartel [109].

As we will see, such a choice of parameters, based on the practice of SELEX / *in vitro* evolution, turns out to be significantly unoptimized. Whereas an accurate selection of the evolutionary scheme, selection, and mutation parameters, allows us to speed up the search hundreds of times and achieve the success rate of 100%. On the contrary, some changes in this initial model of RNA directed evolution can lead to the slowdown of the evolutionary search to its natural limit, the random search.

We will now demonstrate how the effectiveness and success rate of evolutionary search can be increased by varying the parameters of the selection scheme (Table 1). For our example, if

**Table 1. Effectiveness of SELEX tests *in silico*.**

| Search parameters: population size and mutation rate | | # fitness evaluations (st. deviation) | Success rate, % |
|---|---|---|---|
| 200,000 | 1\L | 143 484 447.3 (53 761 568.35) | 70 |
| 1000 | 1\L | 45 547 584.42 (22 087 260.08) | 100 |
| 1000 | 12\L | 1 709 859.03 (663 824.84) | " |
| 1000 | 31\L | 878 326.45 (496 893.84) | " |
| 500 | 31\L | 698 017.25 (417 777.83) | " |
| 250 | 62\L | 532 533.15 (311 239.96) | " |
| 150 | 62\L | 407 487.6 (241 610.77) | " |

Tests are implemented by standard GA with BioRS using $(\mu, \lambda)$ selection, $\mu/\lambda = 0.1$. Averaged over 20 runs. (8 domains, 6 single-valued and 4 double-valued positions out of 26, W = 220, sequence length: $L = (26 + 220) \cdot 8 = 1968$).

we reduce the population size by 200 times, to 1,000, then this will accelerate selection by more than 3 times (from 143 million to ~ 46 million) and increase its success rate to 100% (Table 1).

Further, if we additionally increase the mutation rate by about 12 (or even more) times, then this will further increase the effectiveness of evolutionary search, and it will increase significantly, almost 85 times, as compared with the control (Table 1). The rate increase by 31 times leads to 163-fold decrease of the required number of evaluations. Finally, an increase in the mutation rate up to 1/$NK$ (and even higher) with a decrease in the population size to 150 gives the efficiency for this task close to optimal, so that it becomes 350 (!) times more effective (143 million vs 408 thousand).

The more detailed analysis of the test results from Table 1 is presented in S3 Fig. In particular, we observe the expected property of the evolutionary search that the time necessary to find each next domain grows exponentially (see legend to S3 Fig; Cf [64]). This general property of the evolutionary search with the RS functions (and their extensions) should be borne in mind when planning real experiments with the search from scratch for multidomain artificial macromolecules.

In what follows, we test the effectiveness of the evolutionary search as applied to different versions of our BioRS problem. We start with the simple consensus-based BioRS fitness function. In the standard GA with mutations only we used two selection methods, ($\mu$, $\lambda$) and sigma scaling, which have been shown to be effective for RR problems [19–21]. As a well-known alternative, we tested simple versions of the hill climbing algorithms. This is the random mutation hill-climbing (RMHC) algorithm, the effectiveness of which for RR/RS problems was reported by Mitchell et al. [19–21]. The second algorithm in this family was our version of *parallel RMHC* (see Methods).

Then we address the more realistic competitive version of our BioRS. As it turned out, the competitive BioRS is not only much more complex than its noncompetitive version, though it remains solvable by standard GA it is hardly feasible for RMHC (in contrast to the simple consensus-based BioRS).

## 2.3 Simple consensus-based BioRS: Evolutionary independent domains

Under the assumption that the evolutionary search for the current domain is independent of those already found (noncompetitive BioRS) the search problem gets much easier than in the more general case of the competitive BioRS. Both standard GA and simple RMHC are quite efficient and robust in this case. And above all, they are insensitive to changes of such a key parameter as the mutation rate in the region of its optimal values.

**2.3.1 Search effectiveness is dependent on mutation rate.** Mutation rate is a parameter predominantly affecting the BioRS evolutionary search, therefore we will focus on its contribution to the search effectiveness.

*2.3.1.1 Genetic algorithms*: *Mutation rate*. In comparing algorithms, the main difficulty is that even simple GA with mutations only is controlled by a number of parameters: population size, $\mu$/$\lambda$ selection threshold, mutation rate. We perform the test for a simple consensus based BioRS using the Sassanfar-Szostak motif as a test object. The motif presented in Fig 4E (Structure-I) for computational simplicity is assumed to be a sequence of 10 defined positions out of 26, among which 6 positions (3,5,7,10,11,12) are defined uniquely and 4 positions (4,8,9,13) may take one value out of two possible. For brevity, in what follows we will denote this as "6+4 defined positions". In our specific version of the test (BioRS with 4 equal domains, search window length W = 220), preliminary runs showed that for $\mu$/$\lambda$ = 0.4 and a small population size (600–2000), the efficiency is very high and robust to changes in parameters within the wide range of their values.

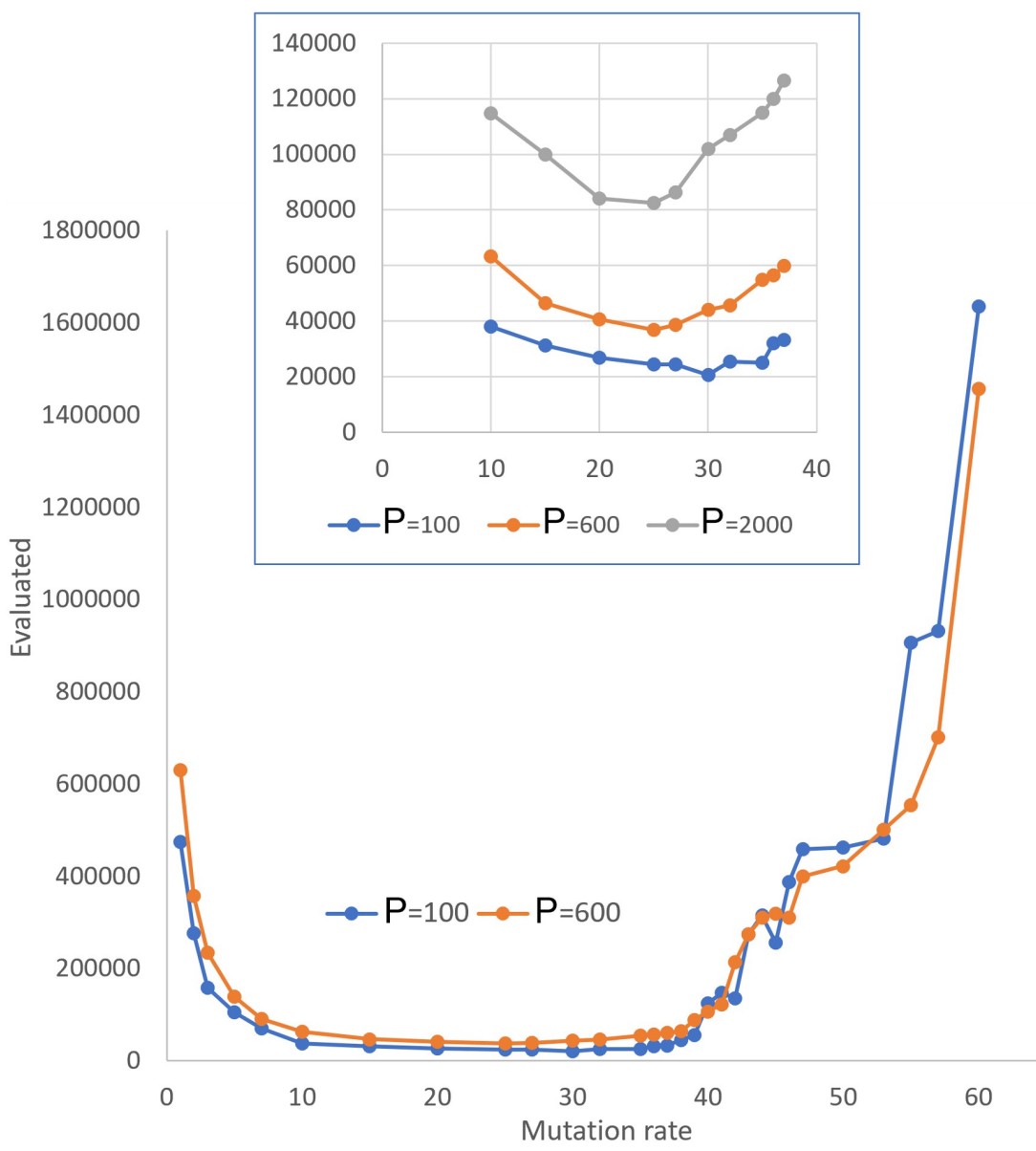

**Fig 7. Effectiveness of evolutionary GA search (mutations only) with ($\mu$, $\lambda$) selection, $\mu/\lambda$ = 0.4, 4 domains, 6+4 defined positions out of 26, W = 220 at varying mutation rate and population size as a function of mutation rate and population size.** In inset the area of the robust algorithm work with low sensitivity to mutation rate is presented at three values of population size: P = 100, 600, and 2000. Within this area populations of low size are preferable. The mean number of fitness evaluations are computed over 100 runs per test. Standard deviations are of value comparable to the mean.

As can be seen from Fig 7, empirical data demonstrate that only at very low and very high values of this parameter the search efficiency is significantly slowed down. In the range from in average 10 mutations per individual to about 35, the search efficiency demonstrates a plateau of values. With a further increase in the mutation rate, the efficiency is initially gradual, and then sharply decreases at a certain critical value of mutation rate. Such a picture is most evident at $\mu/\lambda$ = 0.4 or 0.5. At low $\mu/\lambda$ = 0.05 to 0.1, the efficiency decreases more sharply with an increase in the mutation rate.

Our computational experiments concerning one more important parameter, population size, at varying values of the mutation rate, showed that these two parameters are not independent. In the range of the optimal mutation rate values, the best effectiveness is achieved at lowest possible population sizes. At higher rates, larger populations are required. This effect is demonstrated in Fig 7 in comparison of two graphs computed at different values of population size, 100 vs 600 vs 2000.

*2.3.1.2 Hill Climbing*: *Mutation rate*. RMHC algorithm is much simpler than any version of GA. The main difference is that the search is implemented in one initial sequence instead of a whole population involved in GA. The idea is that after each mutation cycle the fitness of the sequence is computed, and if the fitness is increased, the mutated sequence (offspring) is adopted. Otherwise, the unmutated parent is saved and undergoes the next cycle of mutation. This method is referred to in EC as 1+1 evolution strategy ((1 + 1) EA, e.g. [110]). The scheme of the algorithm is given in Methods.

The dependence of the RMHC efficiency on the mutation rate is comparable to that for the GAs tested in the previous section. The optimal values of mutation rates are very close in these two methods. However, unlike GA, this method does not have a critical speed, above which the search gets extremely ineffective. At high values of the mutation rate, the effectiveness of RMHC also decreases significantly, but not as sharply as in GA. The results of tests applied to the same problem as formulated in «2.3.1.1 Genetic algorithms: mutation rate» are shown in Fig 8.

We have performed the same GA and RMHC tests at other values of parameters and obtained the similar results (not shown here). It is important that in all the cases RMHC is in some extent more efficient than $(\mu, \lambda)$ GA (Fig 8). But even more significant in terms of implementation prospects is that the RMHC algorithm depends on only one parameter and is extremely robust to its changes.

**2.3.2 Parallel RMHC: Effectiveness at parallel processes.** In the parallel version, we run not a single but two or even up to many hundred copies of RMHC in parallel. In this approach, all the once detected domains are preserved in the further search (see Methods). This algorithm seems to be the most promising for *in vitro* implementation. Numerical experiments showed that the search performance is almost comparable to that of a standard RMHC. More specifically, with an increase in the number of such parallel runs, the efficiency gets worse very gradually (almost linearly). The results are shown in Fig 9.

This is important for us in terms of the prospects for implementation in practice because the *in vitro* tests are performed at each moment not by a single RNA molecule (plus its daughter mutant copy), which is easy to lose or destroy, but as much as is reasonable in a particular experiment and without much loss of efficiency. (At 320 parallel runs, for mutagenesis 20 per individual, the efficiency is only two times worse than the optimal one).

**2.3.3 Competitive BioRS: Interdependent domain evolution.** Tests with our rather simple version of the competitive BioRS have led us to some rather discouraging conclusions. We tested the competitive function for an aptamer with alternative consensus (Fig 4E) described in detail in "2.1.2.3 Competitive BioRS function". As it turned out, in the absence of independence between the evolution of individual domains, the patterns of evolutionary search change qualitatively for such problems. At the same time, RMHC completely loses its effectiveness, so we were unable to reach at least the 3rd domain in any of the search runs (for our test problem with 4+10 defined positions out of 26 per domain, W = 220, 4 domains). In GAs this happens because as a result of selection, about half of the individuals with high fitness, for which however the further evolution is impossible, get into the daughter population. More precisely, in order to continue evolution, in this case it is necessary to "return" to lower fitness values and

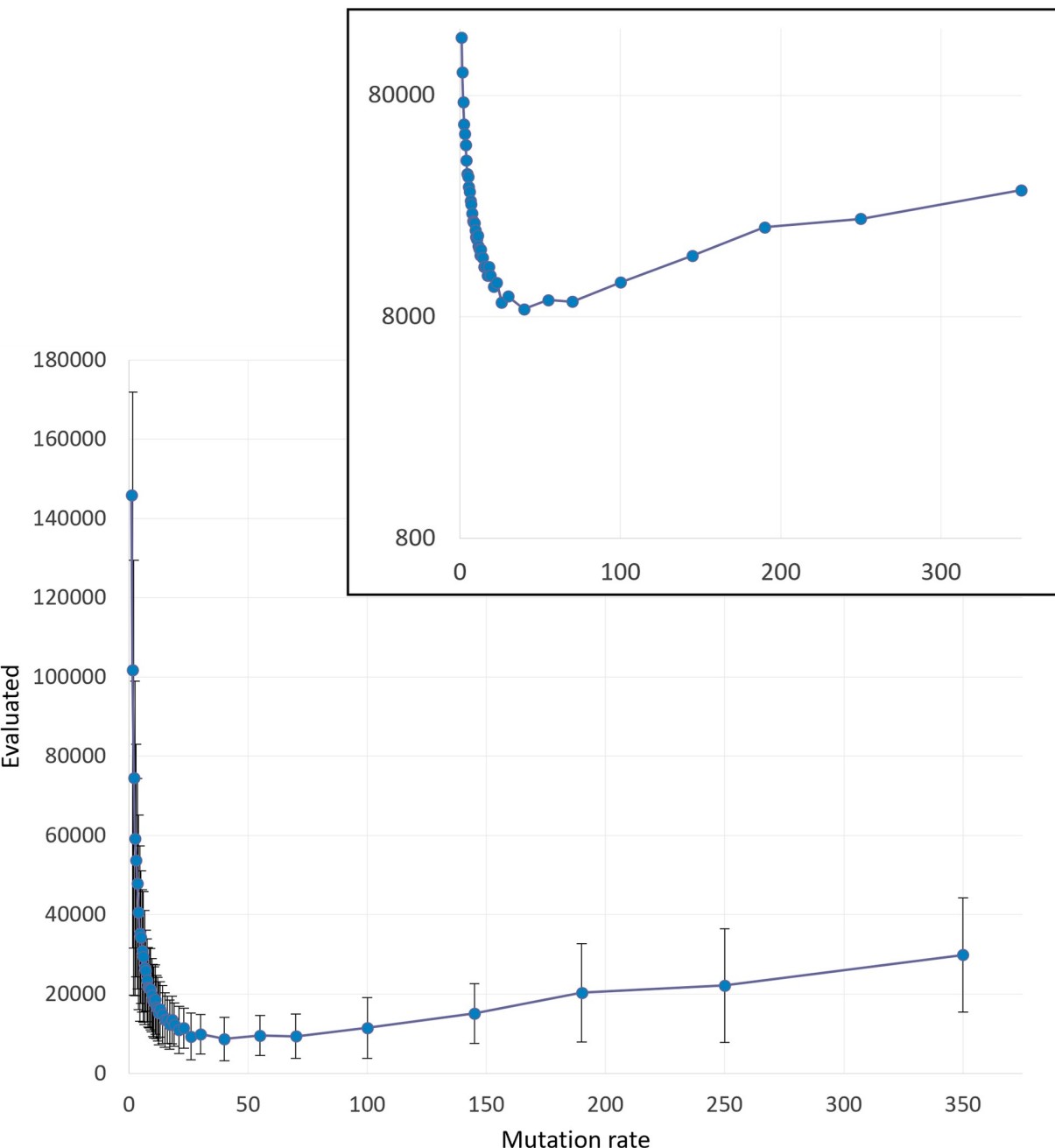

**Fig 8. RMHC: Effectiveness vs mutation rate.** Simple consensus-based BioRS (4 domains, 6+4 defined positions out of 26, W = 220). The mean number of fitness evaluations are computed over 200 runs per test. Standard deviations are shown as vertical bars. Inset is presented in logarithmic coordinates.

find an alternative domain structure (a rod instead of a clover leaf). Obviously, the RMHC does not allow such a "retrograde" movement of the evolutionary search.

Hence, this is the case where we have the opportunity to ensure the importance and necessity of using crossover algorithms to significantly increase the efficiency of evolutionary search.

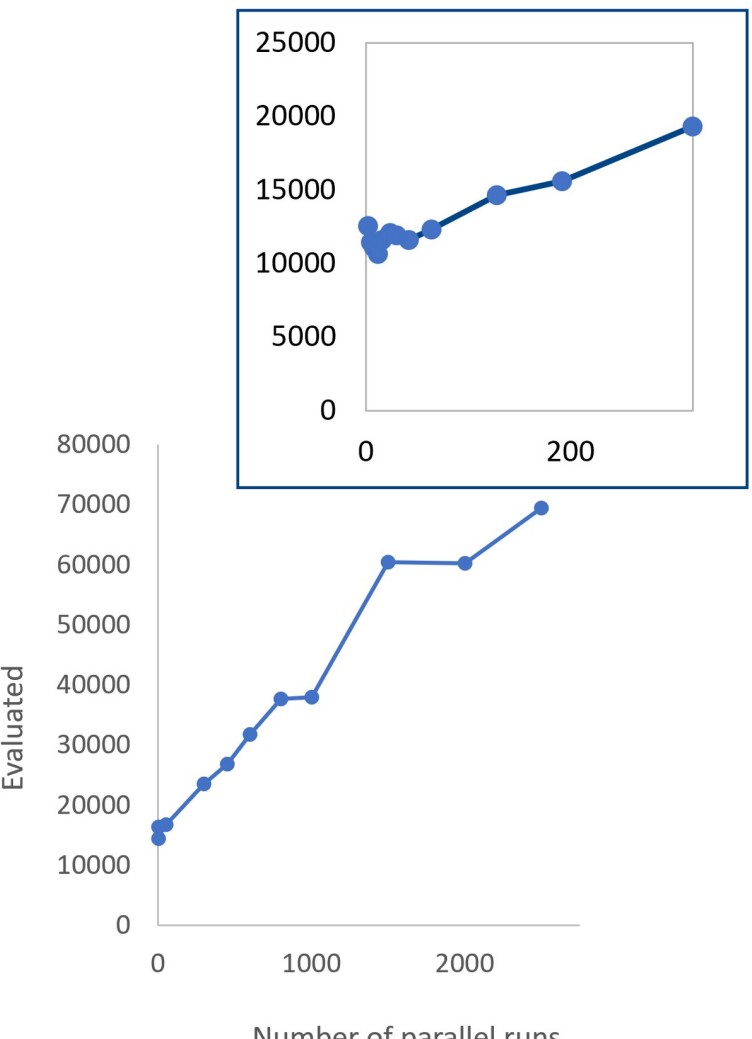

**Fig 9. *In silico* tests with parallel RMHC: Effectiveness vs number of parallel processes.** (4 domains, 6+4 defined positions out of 26, W = 220; mean mutation rate is 20 per chromosome). The mean number of fitness evaluations are computed over 100 runs per test. Standard deviations are of value comparable to the mean.

*2.3.3.1 Competitive BioRS: Simple GAs with crossover.* Holland, Mitchell and Forrest in their classical studies with RR functions found that the "sigma scaling" selection in GA was effective for the RR problem ([111] & our Methods). Therefore, we used this selection method to compare with $(\mu, \lambda)$.

Applying GA with either of these two selection methods did not allow us to achieve 100% success rate even if we do not require the optimal fit to consensus. Results are presented in S1 Table.

Thus, the main problem is that the search with the competitive BioRS demonstrates a low success rate for all the parameter values we tested. The optimal fit to consensus in all four domains (ideal tetramer sequence) is difficult to achieve. While four-domain non-ideal sequences, close enough but not identical to all 4 optimal consensuses, are easier to find. However, the search for the domains with sequences too close to the ideal consensus with the higher fitness level is very resource-intensive.

**Table 2. Effectiveness (number of evaluations) and success rate for the competitive BioRS by means of standard GA with a single-point crossover.**

| Single-point crossover rate | # evaluations (st. deviation) | Success rate (%%) |
|---|---|---|
| 0 | 1 272 681.72 (239 181.20) | 27.5 |
| 0.010 | 1 098 941.33 (134 186.26) | 37.5 |
| 0.025 | 916 979.19 (135 383.02) | 40 |
| 0.035 | 952 473.35 (192 451.17) | 42.5 |
| 0.050 | 806 773.07 (102 602.98) | 35 |

Averaged over 40 runs.

The extensive literature on the problems with RR functions and their extensions leads us to a conclusion about the prospects for using crossover and more advanced crossover-like heuristic algorithms. Therefore, we decided to test some of the simplest versions of the crossover on our test problem. We start with the simplest single-point crossover GA. However, even this technique was enough to increase the efficiency and success rate of our tests. The results are summarized in Table 2.

Tests with a two-point crossover GA algorithm did not show any significant further improvement in efficiency compared to a single-point crossover (results not shown).

*2.3.3.2 Recombination of segments*. We consider the specific versions of BioRS problem, in which it is known where, within what limits, already found and currently searched blocks are located. Such sequence regions are called *segments*. The boundaries of the segments for BioRS are strictly defined (see Fig 3). This allows us to implement our own version of a simple recombination algorithm. Namely, we are implementing a single-point crossover, but such that the position of the point is located exactly at the boundaries of the segments, within which domains are located or searched for. Moreover, such a crossover is only allowed on any of boundaries of the last found segment. Our tests have shown that such a crossover further increases the search speed and success rate, although it does not reach 100%. Moreover, the efficiency of this algorithm depends significantly on the parameters. The results of our tests are summarized in Table 3.

As one can see from Table 3, evolutionary search demonstrates low robustness and when one of the parameters changes, in the general case, it is necessary to pick out the best values of the others by brute force. At the same time, we note that the crossover does not so much accelerate the evolutionary search as it increases the success rate, preventing premature maturation.

Thus, the evolutionary search for competitive BioRS is fundamentally different from the non-competitive one. Here we focus on the questions of the effectiveness of heuristic crossover-like algorithms.

At the same time, the results of this section demonstrate that for about half a dozen or a dozen tests in parallel, it is very likely that at least one of the runs will be successful. If we in one way or another "conjugate" parallel runs in order to prevent premature convergence of the search, then it is very likely that at least some of the parallel runs will be successful, and the search efficiency will be quite high / higher than with separate isolated runs. Therefore, testing and development of parallel GA algorithms looks very promising for such hard problems.

Summarizing the results of our numerical tests with two versions of the BioRS function, simple consensus-based and competitive, we formulate the following conclusions:

1. Unoptimized SELEX may be extremely low effective. Tests with mutation only GA, traditionally used to simulate the standard SELEX, proved that beyond the range of optimal

**Table 3. Effect of the recombination of segments.**

| Analyzed parameter | Values of analyzed parameter | Other parameters | # evaluations (st. deviation) | Success rate,% |
|---|---|---|---|---|
| Mutation rate (Nmut) | 60 | P = 4000, Cros = 0.07 | 781 084.11 (104 616.35) | 22.5 |
| | 120 | " | 1 166 691.6 (141 490.12) | **52.5** |
| | 180 | " | 1 622 052.47 (171 419.97) | 42.5 |
| Crossover rate (Cros) | 0.04 | P = 4000, Nmut = 60 | 1 198 652.53 (104 616.35) | 42.5 |
| | 0.07 | " | 1 176 591.6 (141 500.17) | 52.5 |
| | 0.15 | " | 1 187 205.82 (239 325,97) | 27.5 |
| Population size (P) | 2 000 | Cros = 0.07, Nmut = 60 | **646 037.14** **(67 309.68)** | 35 |
| | 4 000 | " | 1 166 691.6 (141 490.12) | 52.5 |
| | 6 000 | " | 1 639 852.21 (215 441.03) | 47.5 |

Number of evaluations and success rate at different values of the main GA parameters: mutation rate (mean # of mutations), crossover rate, population size. Averaged over 40 runs. The best achieved results are highlighted in bold.

parameter values the search time grows exponentially. Within this range the algorithm performance is robust.

2. Out tests with BioRS functions showed that generation of large populations considerably slows down the search. Large initial populations are only reasonable if they contain a whole domain, sequence of which fits the given consensus. However, for domains of length comparable to real ones, the required size of population is extremely huge. This leads us to use algorithms based on small populations.

3. We tested RMHC algorithms which are known to be effective for RR/RS problems and demonstrated their superiority over standard GAs in application to simple consensus-based BioRS problems. These algorithms do not require population and can be implemented on a single sequence. Our results demonstrate that RMHC provides both higher search speed and stable performance within the wider range of parameter values than GA with large population size.

4. As a most promising to be practically applied we suggest the parallel modification of RMHC algorithm. This version is much more reliable in *in vitro* implementation, and, as our tests have shown, the execution time does not increase significantly with the number of parallel runs.

5. We tested the importance of using crossover to increase the effectiveness of GA evolutionary search for competitive BioRS. Our results demonstrated the promise of further development of this approach in increasing the search speed and success rate.

6. We have also shown that for the competitive BioRS the search effectiveness can be improved using the proposed here specific version of crossover, segment recombination. The prospect of this simple algorithm is that it can be fully implemented in the experiment through already known experimental techniques.

The detailed analysis of our results and conclusions is presented in Discussion.

## 3 Discussion

### 3.1 Domain interference and search from scratch

As we have shown in this article, the problem of finding full-functional multidomain devices from scratch, domain by domain, can be solved by a number of simple heuristic algorithms on small populations and very efficiently. However, this is only so when each next domain can be found with such a structure and position (out of many alternative ones) that it will not interfere with the already found functional domains. In a more general case, the search for the desired multi-domain multifunctional nucleic acid (NA-) devices with a wide range of functions is limited by the interference of the domain that is being currently searched for with those already found. That is, with a given structure of already found functional domains, there may not exist the next domain not interfering with them.

This implicates the obvious advantage of finding multidomain NA-devices from scratch that should not be underestimated. Namely, all other things being equal, the search for several domains from scratch will always be more effective than stitching together already separately found domains available from previous experiments. To increase efficiency, it would be worthwhile to propose a search in several populations in parallel, since this increases the chances of finding combinations of non-interfering domains. Search in the RR paradigm (see Methods), when domains are searched in parallel, can serve the same purposes, especially if it is organized on several populations with known rules of the individual exchange between populations. Therefore, we look at the prospects of searching for NA-devices from scratch with cautious optimism, since we have techniques that allow us to circumvent the general problem of multiple domain interferences in evolutionary search.

### 3.2 Non-optimized SELEX is very lowly effective

GA with the BioRS function (as a model of SELEX) is used to demonstrate that the optimal effectiveness is achieved in the rather wide range of the search parameter values: mutation rate and selection threshold ($\mu/\lambda$). Within this range, the algorithm is quite stable and insensitive to changes in these parameters (Table 1 and Fig 7), while beyond this area, the search time grows exponentially. These results demonstrate how critically important is to optimize the standard SELEX procedures, as well as indicate the ways for finding the preferable parameter values.

Even classical (standard) GAs offer a number of various selection and mutation procedures. Each of these is characterized by its own set of parameters. Therefore, the choice of the most efficient combination of procedures with the best parameter values becomes an essential computational problem. Considering that for situations, anyhow close to real problems of *in vitro* evolution (like our BioRS), computations in time and resources are beyond the limits of the modern computing capabilities, the problem of parameter optimization requires significant efforts.

Moreover, we perform our *in silico* tests in the hope that the found best sets of algorithms with their optimized sets of parameters will be in future implemented as experimental procedures. It is natural to expect that the more parameters a procedure has, the more difficult it is

to implement and the more difficult it is to control it. So simpler techniques should be preferred.

### 3.3 In *in vitro* evolution small populations are preferable

Computational tests with the standard GAs and noncompetitive BioRS function showed that, other things being equal, the effective search is achieved at small populations (Fig 7).

In the standard GA (SELEX), the large populations only make sense if there is a randomly found individuum whose sequence is close enough to the desired (with albeit weak but detectable function). Then the search is directed to the gradual strengthening the function (i.e., increasing the affinity). For the RR/RS problems, this largely loses its sense, since in the search process, we really only possess the information about the fitness function value, i.e. about the number of already found domains, but we have no idea about the quality of the next domain until it is fully identified. Consequently, only a completely defined block can be detected in the initial population, which is very unlikely and requires huge population sizes.

In our BioRS functions, the situation is somewhat closer to experimental due to the assumption of a block redundancy (schemata), i.e., we try to find not a unique sequence but a whole family of those fitting the same (or several) consensus. This somewhat reduces the required size of an initial population but for large domains it still remains huge. The "graduality" we have introduced allows us to improve the affinity of an already found domain, but does not in any way affect the search speed, since the quality of already found domains does not in any way affect the time of finding the next one.

Our test showed that generation of large initial populations considerably slows down the search (Table 1). Within the optimal range of the other parameters (mutation rate and selection threshold μ/λ), the best effectiveness is achieved for small populations. This suggests that other classes of algorithms for solving BioRS problems, not assuming large population size, can be rather promising. As such algorithms, first of all, we see hill climbers which have already shown their effectiveness in solving RR/RS problems.

As it was repeatedly mentioned, the success of the classical SELEX is largely determined by the gigantic number of molecules in the initial population, as well as the simplicity of this experimental approach. New techniques of *in vitro* evolution developed in recent decades have apparent advantages, but are limited to smaller populations. As we have seen in our numerical tests, in order to effectively search from scratch for fully functional multi-domain macromolecules, it is preferable to work with relatively small populations. It means that populations of smaller size than in SELEX are not a disadvantage if the evolutionary search scheme and its parameters are correctly selected. Therefore, our conclusions are relevant primarily for modern cell and microdroplet techniques with next generation sequencing. The main methodological difficulty here is that the evolutionary search will have to be carried out for many tens and even hundreds of thousands generations. With such approaches, the requirements for the reliability of all the experiment stages increase. Search paralleling could help achieve the required level of reliability.

To illustrate the prospects of such techniques we use nucleic acid synthesizers of the CombiMatrix (CustomArray) type as an example. We were inspired to develop this approach by the work of Dr. Kell's group [112–115]. Such devices are capable of synthesizing a large number of RNA (DNA) sequences in parallel, each molecule with a given sequence in a separate "cell". Synthesized populations of identical molecules are tested for the functionality. The next generation sequences are predicted *in silico* by statistical analysis of the better individuals in the current population. Such an approach allows us to specify sequences in the next generation for given mutation, recombination, and selection schemes. Manipulations are performed not

experimentally in test tube but on computer *in silico* [114]. This easily allows us to get the next generation according to the desired schemes and their parameters. In particular, it is possible to mutate and recombine not the whole sequence but just its segment in which the domain we are looking for at this stage (in this epoch) should be located (keeping the already found domains intact). Other heuristic algorithms that are expected to have prospects for experimental implementation are given in a separate subsection. Discussion is below. As far as we know, the main limitations of this methods confine relatively long-time synthesis of sequences along with their length limitations [114]. This is clearly a challenge for experimenters, requiring new technical solutions to implement these laboratory techniques.

## 3.4 Prospects of hill-climbing-based methods

The impressive progress in the field of SELEX / *in vitro* evolution in the 90s of the last century led to the formulation of the rational design methodological paradigm (plug-and-play, as well as its generalizations design-then-select) [116–118]. However, over the past two decades of this century, we have seen how emerging methodological problems and limitations are holding back progress within these paradigms (eg, [46, 118]).

Moreover, even if we have a particular target to make an initially non-working construct stitched from separately functional blocks by evolution into a fully functional one, then such a task is theoretically reduced to a search from scratch for fully functional multi-domain macromolecules.

Progress in the field of directed evolution of nucleic acid (NA-) devices in the last couple of decades has been mainly observed in the field of micro-vesicular and micro-droplet approaches, based on microfluidics and high throughput sequencing. These approaches overcome the fundamental limitations of the classical SELEX [44–46, 118], but the price for this progress is the high complexity and cost of experimental facilities, as well as the critical reduction in the population size that can be worked with. While SELEX allows for populations of $10^{15} - 10^{16}$, these new techniques typically deal with populations of $>10^9$ [44]. Therefore, it is mainly this issue where new methods and approaches are needed to speed up the search and increase its efficiency when dealing with relatively small populations. This is all the more important, since our research has demonstrated that large populations do not provide any advantage in effectiveness of search by heuristic algorithms for BioRS problems. Moreover, for short domains, the much more effective search is achieved on small populations.

As the most promising for practical *in vitro* implementation, we see the parallel modification of the RMHC algorithm not requiring large population. In this case the series of algorithms is run independently during each epoch. The advantage of this approach is that in presence of domain competitiveness, it provides the higher probability of finding domains as non-interfering consensus variants in some of parallel runs. This considerably increases the search effectiveness.

Modern approaches to *in vitro* evolution based on microdroplet technologies in microfluidic devices are best suited to implement our parallel hill climbing algorithms. Such experimental schemes (reviewed in [48, 119, 120]) include at least three microfluidic chips for successive emulsification (1), fusion (2), and sorting (3) activated by fluorescence. Scheme of this experiment is presented in Fig 10. Before being fused, emulsions were collected and thermocycled outside microfluidic chips. After the fusion, they are off-chip incubated in a standard way. Finally, the sorting yields molecules of the improved functionality, if there are any (Fig 10).

The main methodical distinction of our approach is that in each of parallel experiments, manipulations are carried out with a single sequence, more precisely, with a set of identical sequences which is called a *parent population* (Fig 10).

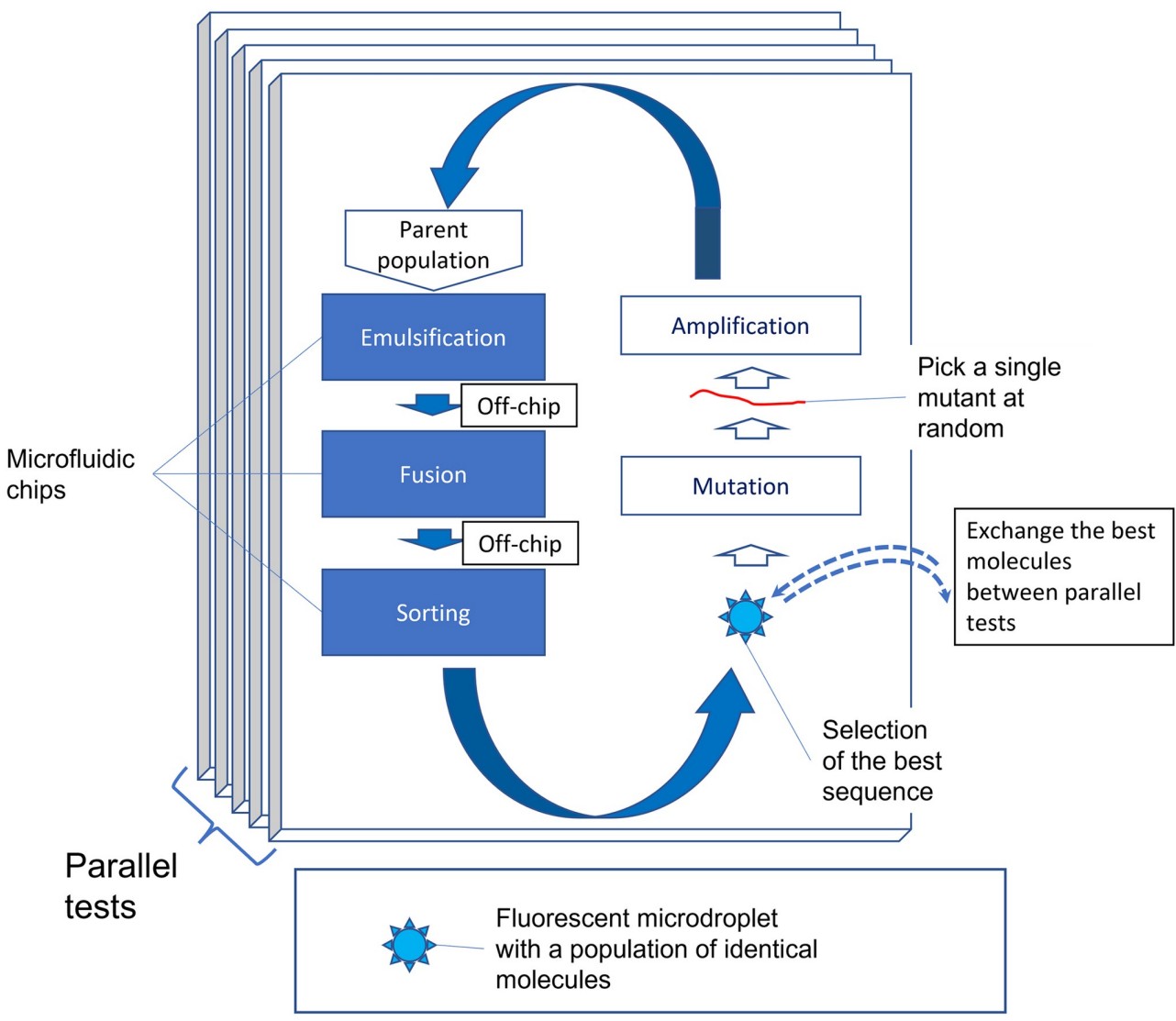

**Fig 10. Block diagram of the experimental setup.** Block diagram of the experimental setup of the experimental setup for the implementation of parallel RMHCs in the experiment. Includes several experiments organized and running in parallel.

At each cycle, each current parent population is subjected to the error-prone PCR to obtain mutant molecules (Fig 10). A few (or even just a single) molecules are picked out of the population and then each of them is used to generate a new population of identical molecules by the standard PCR. This material is emulsified and, finally, the activity of each molecule is evaluated. If the molecule activity is not lower than in the parent population the new population substitutes the previous parent one.

These parallel cycles each of which implementing the hill-climbing are multiply repeated until a desired molecule version is found. Such molecule found in one of the parallels can be used to substitute all or a part of the other parallel populations. Other known parallelization and/or more complex mutation schemes (macromutations in hill-climbing) can also be used (eg, [121, 122]). The results of this article tell us that such an experimental design will demonstrate its effectiveness at sufficiently small population sizes. At the same time, such an

experiment must have a very high reliability for the successful execution of many thousands of cycles. This is undoubtedly a certain challenge to be overcome with a series of new experiments.

## 3.5 Domain interference is a serious problem

Our tests with a competitive version of the BioRS function showed the ineffectiveness of RMHC and an increase in the efficiency of standard GAs when using a crossover (Table 2). We also paid attention to an experimental technique used in *in vitro* evolution for the search of NA-devices. Mutations in this case are only allowed in the new segment within which a new domain is planned to be found, while all the rest, already found, domains are kept intact [54–59]. Following this idea, we have tested the segment recombination algorithm and showed its effectiveness (Table 3). These techniques, the localized segment mutation and recombination, are directly feasible by means of molecular biological procedures, and this is why we consider them rather promising.

Our competitive version of the BioRS function in a simplified form reproduces the problems of evolutionary search for multidomain devices in the presence of interference. Even in such a simple formulation, when only a pair of alternative functional structures and only interference with one nearest domain are taken into account, this evolutionary problem becomes much more difficult. It is natural to expect that assuming multiple interferences between several domains in the course of the search will make the problem even more hard.

In general, such hard problems will require new heuristic approaches to cope with situations like our competitive BioRS, and even much more difficult. Therefore, in this project we are only at the very beginning of the search for new heuristic algorithms that are effective for competitive BioRS and for its more advanced versions.

We are convinced that heuristic algorithms that have demonstrated their efficiency, high success rate and robustness to varying the main critical parameters have good prospects for their implementation in new techniques for directed evolution of macromolecules (in synthetic biology and biotechnology).

In the following final subsection, we will briefly review families of those heuristic algorithms that seem to be promising for the BioRS problems.

## 3.6 New heuristic algorithms for evolution in vitro

In conclusion, we want to draw the reader's attention to the fact that in this publication we tested for efficiency simple classical algorithms effective for RR/RS problems that were developed among the first [19–21, 123]. As it turned out, the effectiveness of a number of these procedures was impressively high. Since then, a wide range of other algorithms have been specifically designed to address RR problems, their extensions, as well as hard functions of other classes [33, 124–128].

The effectiveness of various mutation schemes without crossover on a series of well-known test problems, including RR, was investigated by Richter and Paxton [129]. The mutation scheme according to [130, 131] proved to be the most effective for RR. Further, Watson with coauthors [121, 122] tested HC routines that use the macromutation hill-climber (MMHC) on their RR function extensions, so they should also be tested on BioRS functions.

Over the past decades, many algorithms and approaches have been developed, both quite general and highly specialized, aimed at preserving the BBs [33, 124–128]. First of all, a vast family of such algorithms is made up of non-trivial heuristic recombination algorithms or more general algorithms resembling a crossover.

One of the families of these recombination algorithms, which have clear analogies with the approaches of modern synthetic biology, goes back to the idea of multiple parent recombination

(reviewed in [42, 43]). Such algorithms generate offspring by recombining more than two parental chromosomes, so that, the offspring comes from many fragments from many parents. This is quite reminiscent of the classic DNA shuffling scheme from synthetic biology.

Such algorithms for multi-parental recombination operate in some way with the entire population when obtaining offspring. Therefore, it is possible to preliminarily analyze the population for identifying potential schemata and BBs. Further, such identified blocks can be specially recombined without destroying their integrity. For example, Syswerda's simulated crossover explicitly finds blocks of correlated alleles [132]). This operator has been shown to be effective in a number of tests, including RR functions [133].

Sangkavichitra & Chongstitvatana proposed fairly simple search algorithms that implement the recombination of the found blocks [128, 134]. The efficiency of this algorithm on RR problems is shown in comparison with simple GAs.

BB identification (BBI) methods belong to a specific subarea of the EC (reviewed in [135]). Algorithms using the multivariate interactions model are naturally suitable for the problems considered here, when it is assumed that each block can include several (many) neighboring elements. These are, for example, algorithms such as the Extended compact genetic algorithm (ECGA) and the Bayesian optimization algorithm (BOA) [135]. It is noteworthy that such algorithms resemble some approaches in bioinformatics. First of all, these are algorithms for the identification of autonomous functional domains in protein molecules [16].

Specifically, Mühlenbein and Mahnig [125] have developed two BBI methods (The Univariate Marginal Distribution Algorithm (UMDA) and The Factorized Distribution Algorithm (FDA)) that perform well on optimization problems with RR functions.

Another rather simple version of schema identification by the population analysis with subsequent recombination of the found blocks in the child population was proposed by Kameya and Prayoonsri [33] as the GAP (GA with patterns) algorithm. For the classic version of RR, the GAP algorithm with a two-point crossover is only slightly more efficient than the standard GA with such crossover.

Finally, Ochoa with coauthors proposed Cooperative Co-Evolutionary Algorithms (CCEAs) [127], which proved to be significantly more effective for RR problems than standard GAs.

After the fundamental publications on RR functions, their alternative versions were proposed, which were believed to show better performance of GA, within the framework of the theory of schemata and BBs, in comparison with other heuristic approaches [126]. We are mainly interested in the trap functions, as well as those functions where interactions between blocks were considered and analyzed [121]. The most resounding conclusion from the analysis of such functions is the importance of crossover and crossover-like functions for the effective solution of this kind of hard problems [33, 124–128].

In light of this, it is natural to expect that the next generation algorithms which were for a long time intentionally developed for various optimization problems aimed at preserving the already found BBs, may prove to be significantly, more than an order of magnitude, more efficient than standard GAs. Accordingly, their implementation in experimental procedures of *in vitro* evolution can be very effective and significantly resource-saving. It is in this direction that we plan the further research.

## 4 Methods

### 4.1 Genetic algorithm. Mutation, crossover, and selection methods

**4.1.1 Genetic algorithm (GA).** The objective function in EC is called the fitness function. The basic principles of GA are described by the following scheme:

1. Generate an initial random population of *n* individuals (potential solutions named chromosomes).

2. Calculate its fitness (the value of objective function) for each individual.

3. Select a pair of parent individuals according to their fitness using one of the selection methods.

4. Perform the crossover of two parents with a given probability, producing two offspring.

5. Perform the mutation of offspring with a given probability.

6. Repeat steps 3–5 until a new generation of the population of n individuals is generated.

Repeat steps 2–6 until the optimal value of the fitness function is reached. In the case of GA with mutation only, step 4 is omitted, and all the selected individuals are transferred into the child population.

The main operators of GA are crossover, mutation, and selection into a new population. There are basic forms of operators used to solve specific problems.

Out tests were implemented for two versions of BioRS: simple consensus-based and competitive. The detailed definition of these fitness functions is given below in this section.

In our tests: potential solutions are represented by one-dimensional symbol strings in (A,T, G,C) four-letter alphabet; bitwise mutations, $(\mu, \lambda)$ selection, and single point crossover are applied. Fitness functions for simple consensus-based and competitive BioRS are defined below in this section.

**Mutation** (point mutation) is a random substitution of a sequence symbol by any other from (A,T,G,C) alphabet. Mutations are equally probable in any sequence position; we assign a mean number of mutations per sequence (Nmut) so that the symbol mutation probability, called *mutation rate*, is Nmut/L, where L is the sequence length. This is equivalent to bitwise mutations in standard GAs.

**Crossover (recombination)** is a genetic operator used to combine the information of two parents to generate new offspring. The parents partially exchange their sequences (chromosomes) starting in randomly picked points (single or multiple). This results in two offspring, each carrying some genetic information from both parents.

**4.1.2 ($\mu, \lambda$) selection.**   The parental selection takes place among $\lambda$, the randomly selected and ordered according to their fitness, individuals. The best $\mu$ of these are reproduced with the number of offspring necessary to maintain the population size constant [136]. We usually take $\lambda$ equal to the population size.

**4.1.3 Sigma scaling selection.**   It is a modification of proportional (Roulette Wheel) selection where the number of selected parental individuals is proportional to their fitness. In this version, the number of offspring is normalized according to the variability of fitness across the population. For *i*th individual, the expected number of offspring is $1 + (F_i - -F)/2\sigma$, where $F_i$ is *i*'s fitness, $F$ is the mean fitness of the population, and $\sigma$ is the standard deviation [111, 137].

## 4.2 RMHC and parallel RMHC

**4.2.1 Random-Mutation Hill-Climbing (RMHC) algorithm.**   The algorithm, proposed in the early 90s, is similar to a zero−temperature Metropolis method [20–22, 123]. Given the mean number of mutations per cycle, Nmut, the procedure is as follows:

1. Choose a sequence of length L at random.

2. Mutate each sequence position with the probability Nmut/L.

3. If mutation results in an equal or higher fitness, then the current sequence is replaced by the mutated one.

4. Go to step 2.

5. Repeat 2–4 until the optimal value of the fitness function is reached. Return the current sequence.

**4.2.2 Parallel RMHC.** 1. Choose M sequences of length L at random.

2. Set the threshold fitness value, Δ.

3. For each sequence repeat steps 2–4 of RMHC until fitness of at least one sequence reaches the threshold.

4. Take M copies of the sequence with the highest fitness.

5. Increase the threshold by Δ.

6. Go to 3.

7. Repeat 3–6 until a sequence of fitness NΔ is found.

## 4.3 Royal road functions

The problem with RR fitness function is formulated through the concept of BBs which represent the given short segments of sequence (words). The finding of each of blocks increases the fitness score by an additive value Δ as shown in Fig 11. The value of the RR function at already found K blocks is equal to KΔ. RS function is its modification such that the fitness value is only increased if the BBs are found in a strict order [22]. A more detailed description of the types of RR (RS) functions is presented in S1 File "Royal Road functions".

## 4.4 Biologically substantiated versions of BioRS fitness function

The basic RS fitness functions characterize very simplified structures that can be described by the step function. This function can only take discrete values increasing by a certain value as soon as the next domain (BB) is found. For consensus-based BioRS, there exists a whole set of admissible domain sequences, and as soon as any of them is detected, the fitness function is increased by Δ (we take = 100). Thus, BioRS fitness function preserves its step component registering the finding of a consecutive domain, however, this newly detected domain may additionally contribute into the fitness, hence, instead of the step function, these BioRS functions are gradual, i.e., can take intermediate values.

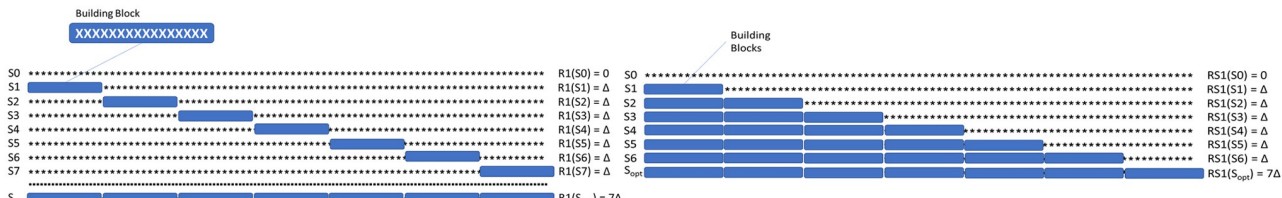

**Fig 11. Schemes of the RR (left) and RS (right) functions.**

Such approach enables us to increase the fitness level of the already found domains in the further search even after the next epoch is achieved. These domains can continue to evolve towards consensus so that they contribute more into the overall fitness of the sequence. Specific implementation of the BioRS versions tested in this study are described below.

**4.4.1 onsensus based BioRS function (simple).** We illustrate the idea using an example of consensus presented in the top of Fig 4E. It is assumed that for the functioning of the desired aptamer we need to detect a certain number of conserved, i.e., uniquely specified, positions in the sequence (specifically, we have 4 positions: 2, 6, 12, and 25). In our particular didactic example, this corresponds to a core "schema" that we can see in two consensus (A) and (C). Whereas the remaining consensus positions are more redundant, and for simplicity, we assume that they can take one value out of two possible (10 positions: 3–5, 7–11, 13, 24). Both values are assumed to fit consensus but only one of them additionally contributes to the sequence fitness (say, the upper values), while the second one is ignored. Each position introduces +1 into the total fitness. If it least one position does not fit the consensus, the total domain fitness is 0 and the block is considered not found. For example, the fitness increment of the sequence N**G**GUG**G**GAACU**G**ANNNNNNNNNNNG**G**N is given by 1+1+0+0+1+0+1+1+1+1 +1+0+1+1 = 10. The total fitness is Δ+10 = 110.

As a final result of the search, all desired domains are found with the sequences rather close to that consensus variant which contributes to the fitness. However, the quality of the domains can be further increased by additional search if we require that the contribution to the overall fitness of each domain be not less than a given value.

**4.4.2 BioRS function with alternative domain structures.** An example is shown in Fig 4E (two lower panels). For simplicity, we assume that each domain has two possible (alternative) structure of approximately the same level of fitness. This approach is a generalization of the consensus based BioRS. The 4 uniquely defined and 10 two-valued positions of the consensus are the same as for the consensus based BioRS shown in the upper panel. Any block fitting this consensus is considered found and the fitness function gains Δ. Next, we adopt a simple model of two alternative structures. 10 two-valued positions are divided into two subsets $m_1$ and $m_2$ (of size 5 + 5), referring to the first and second alternative consensus (two lower panels), respectively. For simplicity, we assume that only those positions contribute into the fitness which coincide with the uniquely defined value of either consensus. For example, only "A" in position 3 of Structure-I may contribute into the fitness, while "G" allowed for Structure-II is ignored. In our example, two subsets are represented by positions $m_1 = (3, 5, 7, 10, 11)$ and $m_2 = (4, 8, 9, 13, 24)$. Then, if positions $m_1$ take specific values uniquely defined in Structure-I (in our example: A, G, G, A, C), or $m_2$ take values uniquely defined in Structure-II (U, C, C, A, C), then the fitness function gains the maximum value (= 5). Conversely, if positions $m_1$ and $m_2$ take alternative values (G, A, A, C, A) and (A, A, A, C, G), respectively, the fitness is equal to 0. All other consensus variants give intermediate fitness increments. We compute fitness increments for both structures and choose the maximum one. For example, the fitness increment of the sequence N**G**GUG**G**GCACU**G**ANNNNNNNNNNNG**G**N is computed for Structure-I as 1+0+0+**1** +1+**1**+0+0+0+0+1+0+0+1 = 6 and for Structure-II as 1+0+**1**+0+**1**+0+**1**+0+0+0+1+**1**+0+1 = 7. Thus, the resulting increment is 7 and the total input of this block into the fitness function is Δ +7.

**4.4.3 Competitive BioRS functions with domain interference.** In this extension of the BioRS function, domains are also assumed to have several variants of consensus, while, not all the combinations of subsequent domains are compatible. A newly detected domain is awarded fitness only under condition that the last found domain corresponds to one specific consensus out of two possible ones, otherwise the fitness of both domains is annihilated, and they are searched anew.

## Supporting information

**S1 Fig. Simple consensus-based BioRS: Series of RMHC tests with the BioRS function at varying W values.**
(PDF)

**S2 Fig. Scheme of a local mutagenesis acting only on the last segment where the search for a new functional domain is performed at the current epoch.**
(PDF)

**S3 Fig. For the BioRS GA the averaged time (in the average, the number of candidate string evaluations) to achieve the (n+1)th fitness level rises exponentially.**
(PDF)

**S1 Table. GA effectiveness for competitive BioRS with (µ,λ) selection.**
(PDF)

**S1 File. Royal road functions.**
(PDF)

**S2 File. Effectiveness of the evolutionary search for problems with variable domain positions.**
(PDF)

## Acknowledgments

We thank the reviewer for his/her valuable comments.

## Author Contributions

**Conceptualization:** Alexander V. Spirov, Ekaterina M. Myasnikova.

**Formal analysis:** Ekaterina M. Myasnikova.

**Funding acquisition:** Alexander V. Spirov.

**Investigation:** Alexander V. Spirov, Ekaterina M. Myasnikova.

**Methodology:** Alexander V. Spirov, Ekaterina M. Myasnikova.

**Project administration:** Alexander V. Spirov.

**Software:** Alexander V. Spirov, Ekaterina M. Myasnikova.

**Validation:** Alexander V. Spirov.

**Visualization:** Alexander V. Spirov.

**Writing – original draft:** Alexander V. Spirov, Ekaterina M. Myasnikova.

**Writing – review & editing:** Ekaterina M. Myasnikova.

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
