## [Decision Letter · Decision Letter 0]

11 Aug 2021

PONE-D-21-21266

Heuristic algorithms in Evolutionary Computations and modular organization of biological macromolecules: applications to in vitro evolution

PLOS ONE

Dear Dr. Spirov,

Thank you for submitting your manuscript to PLOS ONE. After careful consideration, we feel that it has merit but does not fully meet PLOS ONE’s publication criteria as it currently stands. Therefore, we invite you to submit a revised version of the manuscript that addresses the points raised during the review process.

We look forward to receiving your revised manuscript.

Kind regards,

Josh Bongard

Academic Editor

PLOS ONE

Journal Requirements:

 [The funders had no role in study design, data collection and analysis, decision to publish, or preparation of the manuscript]. 

[This work is supported by the Russian Foundation for Basic Research (grant no. 20-04-01015) and Federal Agency for Scientific Organizations of the Russian Federation (grant no. 01201351572).]

 [The funders had no role in study design, data collection and analysis, decision to publish, or preparation of the manuscript.]

Reviewers' comments:

Reviewer's Responses to Questions

**Comments to the Author**

1. Is the manuscript technically sound, and do the data support the conclusions?

Reviewer #1: Partly

2. Has the statistical analysis been performed appropriately and rigorously? 

Reviewer #1: No

3. Have the authors made all data underlying the findings in their manuscript fully available?

Reviewer #1: Yes

4. Is the manuscript presented in an intelligible fashion and written in standard English?

Reviewer #1: No

5. Review Comments to the Author

Reviewer #1: It is difficult to determine if the work is technically sound as the authors need

to explain more clearly and specically what their motivations and goals were

for this research, or what questions they were looking to answer. How their

methods were implemented in silico was not explained clearly, and they do

not summarize what specic conclusions they have drawn from their results.

For example the authors reference the SELEX method, an existing in vitro

method for evolution DNA aptamers. What are they specically hoping to

contribute to methods such as SELEX?

In the methods section they summarize various evolutionary algorithms,

but they do not explain how the authors themselves explicitly applied them

in their experiments. It is also not clear from the methods section how these

various algorithms which they discuss related to their research questions or

goals.

The authors should dene clearly in their introduction what questions

they are seeking to answer with this research, what their novel contributions

were, and what their utility is.

The results section is very verbose and contains a signiant discussion

of related work, so it is difficult to determine what questions the various

experiments or case studies were seeking to answer and whether they did so

successfully. The authors do not clearly dene what success means in their

various experiments, and they need to explain more clearly what the exact

tness function was, how it was evaluated, and how it was implemented in

silico. It is therefore difficult to identify what conclusions can be drawn from

the paper.

It was only evident after reading the entire results section that the point

of the in silico experiments was to identify the optimal algorithm features

and hyperparameters for in vitro implementation through in silico simulation.

This should be made clear in the introduction.

The discussion also contains a large discussion of related work rather

than a discussion of their experimental results, and so after reading it, it is

still unclear what conclusions the authors have drawn from the research, or

whether their conclusions are novel, and what their utility is.

The authors need a separate section where they summarize their conclusions,

to discuss how their results lead to those conclusions, and connect their

conclusions back to the goals or questions which they were hoping to answer

with this research.

The experiments conducted in the paper were all in silico simulations. These

simulations were composed of an analysis of their proposed evolutionary

algorithm from different perspectives. Therefore, the expected statistical analysis

would involve typical metrics often reported when analysing the performance

of evolutionary algorithms.

When analysing the performance of evolutionary algorithms one generally

needs to report:

-Some variation of average success rate, that is, the fraction of algorithm

runs where a solution of satisfactory quality was obtained.

-The mean best tness, which is the optimal tness obtained from each

run averaged over the number of runs.

-The effectiveness, or the average solution quality obtainable within a

given computational limit.

-Or the efficiency, which is the average amount of time needed to achieve

a solution of satisfactory quality. Given that computational time depends

on the available hardware, efficiency is often reported as the

average number of evaluations to a solution.

These are also usually accompanied by progress curves for the various

metrics. The authors reported success rate, effectiveness and efficiency in

various gures throughout the results section. These reported metrics are

sufficient for the nature of their in silico experiments, however, it is not

always made clear how success was dened, that is, how the authors dened

a solution to be of acceptable quality.

There are many grammatical problems with the text, potentially because the

author's rst language is not English. So much so that it makes it difficult

to understand what the authors are trying to communicate in many places.

Further, some concepts were not sufficiently claried. The following concepts

need to be explained more clearly:

-Directed evolution / articial evolution of biological macromolecules.

-How evolution of biological macromolecules is directed in vitro.

-In the case studies, how were the molecules which were being evolved,

represented and evaluated in silico.

-Implementation details of the selection, mutation and crossover /

recombination methods.

Regarding the layout of the paper, the research questions or goals of the

research need to be clearly stated in the introduction. The introduction

should also clearly describe what this research contributes to the field.

Discussions of related work should be separated from the results and discussion

sections and moved to a separate related work section. The results section

should clearly explain what experiments were run and what results were ob-

tained from each experiment. The discussion section should discuss these

results, not related work, and extract conclusions from them. The conclu-

sions section should summarize everything which can be extracted from the

results and connect the conclusions back to the goals and questions of the

research.

6. PLOS authors have the option to publish the peer review history of their article (what does this mean?). If published, this will include your full peer review and any attached files.

Reviewer #1: No

---

## [Author Response · Author response to Decision Letter 0]

2 Nov 2021

Dear Dr Bongard,

Thank you for the letter, regarding manuscript, ‘Heuristic algorithms in Evolutionary Computations and modular organization of biological macromolecules: applications to in vitro evolution ' by Spirov & Myasnikova. We hereby resubmit the manuscript. 

Above all we would like to extend our thanks and appreciation to the editors and reviewer for making very important and helpful comments. 

We modified the manuscript according to the reviewer’s recommendations. The major corrections are highlighted in red. Our point-by-point response to the reviewer’s comments follows. For the sake of convenience, the comments are enumerated. The page numbers are referred to according to the revised manuscript. 

The manuscript was strongly revised, structured, more illustrations (two new figures), and references were added. Several figures were revised. In response to the Reviewer’s comments, Introduction and Discussion were restructured, Results section was shortened. The conclusions from our numerical tests were explicitly formulated. 

The goal and major tasks of our study are now explicitly formulated in Introduction. 

The most significant insertions have been introduced into Discussion. The reviewer drew our attention to the fact that the prospects for our research were not sufficiently elucidated`. In response to this comment, we have added extended paragraphs discussing the prospects for experimental implementation of the algorithms we have proposed and tested.

We hope to have given the full responses to all the comments. 

Thank you for your consideration,

Sincerely,

Alexander Spirov

Ekaterina Myasnikova

 

PLOS ONE

Journal Requirements:

Done. 

 [The funders had no role in study design, data collection and analysis, decision to publish, or preparation of the manuscript]. 

The research was supported by the Russian Science Foundation (grant 17-18-01536). 

[This work is supported by the Russian Foundation for Basic Research (grant no. 20-04-01015) and Federal Agency for Scientific Organizations of the Russian Federation (grant no. 01201351572).]

 [The funders had no role in study design, data collection and analysis, decision to publish, or preparation of the manuscript.]

Done. 

Done. 

Done.

Reviewers' comments:

Reviewer #1:

1. It is difficult to determine if the work is technically sound as the authors need

to explain more clearly and specically what their motivations and goals were

for this research, or what questions they were looking to answer. 

Our motivation: we, like some other researchers, are inspired by the similarity of ideas in the field of evolutionary computation and in vitro evolution, in general, and SELEX, in particular. We emphasis this in Introduction (1.1 Biological in vitro evolution and genetic algorithms; 1.2 Building Blocks in biology and genetic algorithms) and Discussion. Moreover, we are inspired by our observation that well-known in GA foundations Royal Roads problem is very similar to the problem of evolutionary search from scratch for multidomain (and multifunctional) synthetic macromolecules. This remarkable similarity inspired us for this study. In Discussion we also discuss the ways and prospects of implementation of more effective algorithms into biological experiment that can be also regarded as our motivation. 

In order to convey our motives to potential readers, we have added a new first figure (Fig. 1) to Introduction. This figure illustrates our expectation that "Heuristic algorithms from evolutionary computation hold remarkable promise for transfer to synthetic biology and biotechnology." 

We have clearly formulated the goal and two major tasks of the article (last para in Introduction, p. 7). 

2. How their methods were implemented in silico was not explained clearly, and they do

not summarize what specific conclusions they have drawn from their results.

We expanded the description of the methods and their specific implementation in the Methods section. Specifically, we added the details of our implementation of mutations and crossover. 

The detailed description of standard selection methods ((µ, λ) and Sigma scaling selection) and specific versions of BioRS (biologically substantiated versions of BioRS fitness function) was previously included into Methods. 

To bring the BioRS function notations into correspondence with the rest of the text, the last subsection is now entitled “Competitive BioRS function (with domain interference)” and the explanation: “This version of the BioRS function is a further extension of “BioRS function with alternative domain structures” in which the domain interference is additionally taken into account” is introduced. 

To make the application of these methods clearer for readers, we have introduced several relevant links to this section from those places where the corresponding method is used. 

We have formulated the main conclusions summarizing the results of our numerical tests in the end of the Results section (pp. 28-29). We have also significantly expanded and restructured the Discussion section where these conclusions are analyzed and discussed (see also our responses below). 

3. For example the authors reference the SELEX method, an existing in vitro

method for evolution DNA aptamers. What are they specifically hoping to

contribute to methods such as SELEX?

We have added extended paragraphs to Discussion, where we formulate the prospects of experimental implementation of the algorithms we have tested. Specifically, we discuss modern experimental approaches of in vitro evolution using nucleic acid synthesizers, as it was done by prof. Kell’s group [Knight et al., 2009] (3.3 “In in vitro evolution small populations are preferable”, final paragraph (p.31)), and microfluidic techniques (3.4 “Prospects of hill-climbing-based methods” final paragraphs (pp.32-33)). 

4. In the methods section they summarize various evolutionary algorithms,

but they do not explain how the authors themselves explicitly applied them

in their experiments. It is also not clear from the methods section how these

various algorithms which they discuss related to their research questions or

goals.

The Methods section only encompasses those algorithms that are tested in our study. Now we specify these algorithms explicitly: “Our tests were implemented for two versions of BioRS: simple consensus-based and competitive.” To avoid the confusion the notation of the BioRS function with domain interference is corrected to “Competitive BioRS with domain interference”. See also the response to the previous comment. 

Our initial selection of (new) heuristic algorithms for in silico testing was inspired and dictated by the RR research area in GA. It was there that the standard (mu, lambda)-GA was tested in comparison with Sigma scaling selection, and in comparison with RMHC. We emphasize this in the text (See, for example, 2.2, 2nd last para, p.22). To develop this issue, we have also proposed and tested a parallel RMHC. A more complete list of promising algorithms that we have not tested yet, but which have shown their effectiveness in GA for RR problems, is given in Discussion. We expect that algorithms that are effective for RR problems in GA may turn out to be promising for in vitro evolution problems (which we repeatedly emphasize in the text). We have restructured the Results section to make it clearer for the reader. We have also paid attention to this issue in Discussion (final subsection). 

5. The authors should define clearly in their introduction what questions

they are seeking to answer with this research, what their novel contributions

were, and what their utility is.

The Introduction section has been substantially modified and structured to clearly outline our motives, our goal and the tasks we solve, as well as our most important result. As the most significant new result, we interpret our conclusion about the prospects of implementing the RMHC & parallel RMHC algorithms in new experimental approaches. 

6. The results section is very verbose and contains a signiant discussion

of related work, so it is difficult to determine what questions the various

experiments or case studies were seeking to answer and whether they did so

successfully. The authors do not clearly dene what success means in their

various experiments, and they need to explain more clearly what the exact

tness function was, how it was evaluated, and how it was implemented in

silico. It is therefore difficult to identify what conclusions can be drawn from

the paper.

We have somewhat shortened or completely removed to Discussion the introductory paragraphs in Results subsections to present our work and its results more clearly. We explicitly presented our fitness function in Methods and explained how it was evaluated, and how it was implemented in silico. We have added key conclusions at the end of the Results (pp. 28-29) and discussed them in detail in the new Discussion subsections (see below). 

7. It was only evident after reading the entire results section that the point

of the in silico experiments was to identify the optimal algorithm features

and hyperparameters for in vitro implementation through in silico simulation.

This should be made clear in the introduction.

Our intentions, goals and objectives are now clearly and explicitly formulated in the end of Introduction, p.7 (as noted above). 

8. The discussion also contains a large discussion of related work rather

than a discussion of their experimental results, and so after reading it, it is

still unclear what conclusions the authors have drawn from the research, or

whether their conclusions are novel, and what their utility is.

The authors need a separate section where they summarize their conclusions,

to discuss how their results lead to those conclusions, and connect their

conclusions back to the goals or questions which they were hoping to answer

with this research.

As already mentioned above, we substantially supplemented and structured Discussion to discuss the results of our numerical tests and conclusions from them. The novelty of our research is determined by the novel idea of introducing the biologically substantiated extensions of RS-type fitness function that have prospects in experimental applications. This is explained in Introduction. 

9. The experiments conducted in the paper were all in silico simulations. These

simulations were composed of an analysis of their proposed evolutionary

algorithm from different perspectives. Therefore, the expected statistical analysis

would involve typical metrics often reported when analysing the performance

of evolutionary algorithms.

When analyzing the performance of evolutionary algorithms one generally

needs to report:

-Some variation of average success rate, that is, the fraction of algorithm

runs where a solution of satisfactory quality was obtained.

-The mean best fitness, which is the optimal fitness obtained from each

run averaged over the number of runs.

-The effectiveness, or the average solution quality obtainable within a

given computational limit.

-Or the efficiency, which is the average amount of time needed to achieve

a solution of satisfactory quality. Given that computational time depends

on the available hardware, efficiency is often reported as the

average number of evaluations to a solution.

These are also usually accompanied by progress curves for the various

metrics. The authors reported success rate, effectiveness and efficiency in

various figures throughout the results section. These reported metrics are

sufficient for the nature of their in silico experiments, however, it is not

always made clear how success was defined, that is, how the authors defined

a solution to be of acceptable quality.

We have completed all the Tables, as well as the corresponding figures (Fig. 7; 8; 9) so that all the test criteria adopted in the GA are given explicitly. Specifically, when comparing algorithms, we give the average number of fitness evaluations, standard deviation, and success rate. 

Additionally, we present in Supplement and briefly analyze the graphs presenting the mean duration (# of evaluations) of each next epoch in GA tests at different parameter values. These tests (Table 1) were implemented for the specific BioRS version with eight domains to make this empirical analysis more illustrative. 

10. There are many grammatical problems with the text, potentially because the

author's rst language is not English. So much so that it makes it difficult

to understand what the authors are trying to communicate in many places.

The spell check is done, grammar is corrected. 

11. Further, some concepts were not sufficiently claried. The following concepts

need to be explained more clearly:

-Directed evolution / artificial evolution of biological macromolecules. 

The definition is given by an endnote. 

-How evolution of biological macromolecules is directed in vitro.

This same endnote also provides a response to this comment. 

-In the case studies, how were the molecules which were being evolved,

represented and evaluated in silico. 

As we explained now in the Methods section, in our tests: potential solutions are represented by one-dimensional symbol strings in (A,T,G,C) four-letter alphabet; bitwise mutations, (µ, λ) selection, and single point crossover are applied. Fitness functions for simple consensus-based and competitive BioRS are defined below in this section (p.36). 

-Implementation details of the selection, mutation and crossover /

recombination methods.

Implementation details are now given in Methods (pp.36-37). 

12. Regarding the layout of the paper, the research questions or goals of the

research need to be clearly stated in the introduction. The introduction

should also clearly describe what this research contributes to the field.

Done. Please see our responses above. 

13. Discussions of related work should be separated from the results and discussion

sections and moved to a separate related work section. The results section

should clearly explain what experiments were run and what results were ob-

tained from each experiment. The discussion section should discuss these

results, not related work, and extract conclusions from them. The conclu-

sions section should summarize everything which can be extracted from the

results and connect the conclusions back to the goals and questions of the

research.

As discussed above, the main sections of the article, in particular Discussion, are modified, restructured in response to the Reviewer’s comments. 

6. PLOS authors have the option to publish the peer review history of their article (what does this mean?). If published, this will include your full peer review and any attached files.

---

## [Decision Letter · Decision Letter 1]

11 Nov 2021

Heuristic algorithms in Evolutionary Computations and modular organization of biological macromolecules: applications to in vitro evolution

PONE-D-21-21266R1

Dear Dr. Spirov,

We’re pleased to inform you that your manuscript has been judged scientifically suitable for publication and will be formally accepted for publication once it meets all outstanding technical requirements.

Kind regards,

Josh Bongard

Academic Editor

PLOS ONE

Additional Editor Comments (optional):

Reviewers' comments:

Reviewer's Responses to Questions

**Comments to the Author**

1. If the authors have adequately addressed your comments raised in a previous round of review and you feel that this manuscript is now acceptable for publication, you may indicate that here to bypass the “Comments to the Author” section, enter your conflict of interest statement in the “Confidential to Editor” section, and submit your "Accept" recommendation.

Reviewer #1: All comments have been addressed

2. Is the manuscript technically sound, and do the data support the conclusions?

Reviewer #1: Yes

3. Has the statistical analysis been performed appropriately and rigorously? 

Reviewer #1: Yes

4. Have the authors made all data underlying the findings in their manuscript fully available?

Reviewer #1: Yes

5. Is the manuscript presented in an intelligible fashion and written in standard English?

Reviewer #1: Yes

6. Review Comments to the Author

Reviewer #1: The authors have satisfactorily addressed all comments and concerns.

7. PLOS authors have the option to publish the peer review history of their article (what does this mean?). If published, this will include your full peer review and any attached files.

Reviewer #1: No

---

## [Editor Report · Acceptance letter]

22 Nov 2021

PONE-D-21-21266R1 

Heuristic algorithms in Evolutionary Computations and modular organization of biological macromolecules: applications to in vitro evolution 

Dear Dr. Spirov:

I'm pleased to inform you that your manuscript has been deemed suitable for publication in PLOS ONE. Congratulations! Your manuscript is now with our production department. 

Kind regards, 

on behalf of

Dr. Josh Bongard 

Academic Editor

PLOS ONE